# A Trichotomy for Transductive Online Learning

**Steve Hanneke**
Department of Computer Science
Purdue University
steve.hanneke@gmail.com

**Shay Moran**
Faculty of Mathematics,
Faculty of Computer Science, and
Faculty of Data and Decision Sciences
Technion – Israel Institute of Technology
smoran@technion.ac.il

**Jonathan Shafer**
Computer Science Division
UC Berkeley
shaferjo@berkeley.edu

## Abstract

We present new upper and lower bounds on the number of learner mistakes in the 'transductive' online learning setting of Ben-David, Kushilevitz and Mansour (1997). This setting is similar to standard online learning, except that the adversary fixes a sequence of instances $x_1, \ldots, x_n$ to be labeled at the start of the game, and this sequence is known to the learner. Qualitatively, we prove a *trichotomy*, stating that the minimal number of mistakes made by the learner as $n$ grows can take only one of precisely three possible values: $n$, $\Theta\left(\log(n)\right)$, or $\Theta(1)$. Furthermore, this behavior is determined by a combination of the VC dimension and the Littlestone dimension. Quantitatively, we show a variety of bounds relating the number of mistakes to well-known combinatorial dimensions. In particular, we improve the known lower bound on the constant in the $\Theta(1)$ case from $\Omega\left(\sqrt{\log(d)}\right)$ to $\Omega(\log(d))$ where $d$ is the Littlestone dimension. Finally, we extend our results to cover multiclass classification and the agnostic setting.

## 1 Introduction

In classification tasks like PAC learning and online learning, the learner simultaneously confronts two distinct types of uncertainty: *labeling-related* uncertainty regarding the best labeling function, and *instance-related* uncertainty regarding the instances that the learner will be required to classify in the future. To gain insight into the role played by each type of uncertainty, researchers have studied modified classification tasks in which the learner faces only one type of uncertainty, while the other type has been removed.

In the context of PAC learning, [BB11] studied a variant of proper PAC learning in which the true labeling function is known to the learner, and only the distribution over the instances is not known. They show bounds on the sample complexity in this setting, which conceptually quantify the instance-related uncertainty. Conversely, labeling-related uncertainty is captured by PAC learning with respect to a fixed (e.g., uniform) domain distribution [BI91], a setting which has been studied extensively.

In this paper we improve upon the work of [BKM97], who quantified the label-related uncertainty in online learning. They introduced a model of *transductive online learning*,[1] in which the adversary

---

[1][BKM97] call their model 'off-line learning with the worst sequence', but in this paper we opt for 'transductive online learning', a name that has appeared in a number of publications, including [KK05, Pec08, CS13, SKS16]. We remark there are at least two different variants referred to in the litera-

commits in advance to a specific sequence of instances, thereby eliminating the instance-related uncertainty.

## 1.1 The Transductive Online Learning Model

The model of learning studied in this paper, due to [BKM97], is a zero-sum, finite, complete-information, sequential game involving two players, the *learner* and the *adversary*. Let $n \in \mathbb{N}$, let $\mathcal{X}$ and $\mathcal{Y}$ be sets, and let $\mathcal{H} \subseteq \mathcal{Y}^{\mathcal{X}}$ be a collection of functions called the *hypothesis class*.

The game proceeds as follows (see Section 2 for further formal definitions). First, the adversary selects an arbitrary sequence of *instances*, $x_1, \ldots, x_n \in \mathcal{X}$. Then, for each $t = 1, \ldots, n$:

1. The learner selects a *prediction*, $\hat{y}_t \in \mathcal{Y}$.

2. The adversary selects a *label*, $y_t \in \mathcal{Y}$.

In each step $t \in [n]$, the adversary must select a label $y_t$ such that the sequence $(x_1, y_1), \ldots, (x_t, y_t)$ is *realizable* by $\mathcal{H}$, meaning that there exists $h \in \mathcal{H}$ satisfying $h(x_i) = y_i$ for all $i \in [t]$. The learner's objective is to minimize the quantity

$$M(A, x, h) = |\{t \in [n] : \hat{y}_t \neq h(x_t)\}|,$$

which is the number of mistakes when the learner plays strategy $A$ and the adversary chooses sequence $x \in \mathcal{X}^n$ and labels consistent with hypothesis $h \in \mathcal{H}$. We are interested in understanding the value of this game,

$$M(\mathcal{H}, n) = \inf_{A \in \mathcal{A}} \sup_{x \in \mathcal{X}^n} \sup_{h \in \mathcal{H}} M(A, x, h),$$

where $\mathcal{A}$ is the set of all learner strategies. Note that neither party can benefit from using randomness, so without loss of generality we consider only deterministic strategies.

## 1.2 A Motivating Example

Transductive predictions of the type studied in this paper appear in many real-world situations, essentially in any case where a party has a schedule or a to-do list known in advance of specific tasks that need to be completed in order, and there is some uncertainly as to the precise conditions that will arise in each task. As a concrete example, consider the logistics that the management of an airport faces when scheduling the work of passport-control officers.

**Example 1.1.** *An airport knows in advance what flights are scheduled for each day. However, it does not know in advance exactly how many passengers will go through passport control each day (because tickets can be booked and cancelled in the last minute, and entire flights can be cancelled, delayed or rerouted, etc.). Each day can be 'regular' (the number of passengers is normal), or it can be 'busy' (more passengers than usual). Correspondingly, each day the airport must decide whether to schedule a 'regular shift' of passport-control officers, or an 'extended shift' that contains more officers.*

*If the airport assigns a standard shift for a busy day, then passengers experience long lines at passport control, and the airport suffers a loss of 1; if the airport assigns an extended shift for a regular day, then it wastes money on excess manpower, and it again experiences a loss of 1; If the airport assigns a regular shift to a regular day, or an extended shift to a busy day, then it experiences a loss of 0.*

Hence, when the airport schedules its staff, it is essentially attempting to predict for each day whether it will be a regular day or a busy day, using information it knows well in advance about which flights are scheduled for each day. This is precisely a transductive online learning problem.

## 1.3 Our Contributions

**I. Trichotomy.** We show the following *trichotomy*. It shows that the rate at which $M(\mathcal{H}, n)$ grows as a function of $n$ is determined by the VC dimension and the Littlestone dimension (LD).

---

ture as 'transductive online learning'. For example, [SKS16] write of "a transductive setting [BKM97] in which the learner knows the arriving contexts a priori, or, less stringently, knows only the set, but not necessarily the actual sequence or multiplicity with which each context arrives." That is, in one setting, the learner knows the sequence $(x_1, \ldots, x_n)$ in advance, but in another setting the learner only knows the set $\{x_1, \ldots, x_n\}$. One could distinguish between these two settings by calling them 'sequence-transductive' and 'set-transductive', respectively. Seeing as the current paper deals exclusively with the sequence-transductive setting, we refer to it herein simply as the 'transductive' setting.

**Theorem** (**Informal Version of Theorem 4.1**). *Every hypothesis class $\mathcal{H} \subseteq \{0,1\}^{\mathcal{X}}$ satisfies precisely one of the following:*

1. *$M(\mathcal{H}, n) = n$. This happens if $\mathsf{VC}(\mathcal{H}) = \infty$.*

2. *$M(\mathcal{H}, n) = \Theta(\log(n))$. This happens if $\mathsf{VC}(\mathcal{H}) < \infty$ and $\mathsf{LD}(\mathcal{H}) = \infty$.*

3. *$M(\mathcal{H}, n) = \Theta(1)$. This happens if $\mathsf{LD}(\mathcal{H}) < \infty$.*

*The $\Theta(\cdot)$ notations in Items 2. and 3. hide a dependence on $\mathsf{VC}(\mathcal{H})$, and $\mathsf{LD}(\mathcal{H})$, respectively.*

The proof uses bounds on the number of mistakes in terms of the *threshold dimension* (Section 3.2), among other tools.

II. **Littlestone classes.** The minimal constant upper bound in the $\Theta(1)$ case of Theorem 4.1 is some value $C(\mathcal{H})$ that depends on the class $\mathcal{H}$, but the precise mapping $\mathcal{H} \mapsto C(\mathcal{H})$ is not known in general. [BKM97] showed that $C(\mathcal{H}) = \Omega\left(\sqrt{\log(\mathsf{LD}(\mathcal{H}))}\right)$. In Section 3 and Appendix A we improve upon their result as follows.

**Theorem** (**Informal Version of Theorem 3.1**). *Let $\mathcal{H} \subseteq \{0,1\}^{\mathcal{X}}$ such that $\mathsf{LD}(\mathcal{H}) = d < \infty$. Then $M(\mathcal{H}, n) = \Omega(\log(d))$.*

III. **Multiclass setting.** In Section 5, we generalize Theorem 4.1 to the multiclass setting with a finite label set $\mathcal{Y}$, showing a trichotomy based on the Natarajan dimension. The proof uses a simple result from Ramsey theory, among other tools.

Additionally, we show that the DS dimension of [DS14] does not characterize multiclass transductive online learning.

IV. **Agnostic setting.** In the *standard* (non-transductive) agnostic online setting, [BPS09] showed that $R_{\text{online}}(\mathcal{H}, n)$, the agnostic online regret for a hypothesis class $\mathcal{H}$ for a sequence of length $n$ satisfies

$$\Omega\left(\sqrt{\mathsf{LD}(\mathcal{H}) \cdot n}\right) \leq R_{\text{online}}(\mathcal{H}, n) \leq O\left(\sqrt{\mathsf{LD}(\mathcal{H}) \cdot n \cdot \log n}\right). \tag{1}$$

Later, [ABD$^+$21] showed an improved bound of $R_{\text{online}}(\mathcal{H}, n) = \Theta\left(\sqrt{\mathsf{LD}(\mathcal{H}) \cdot n}\right)$.

In Section 6 we show a result similar to Eq. (1), for the *transductive* agnostic online setting.

**Theorem** (**Informl Version of Theorem 6.1**). *Let $\mathcal{H} \subseteq \{0,1\}^{\mathcal{X}}$, such that $0 < \mathsf{VC}(\mathcal{H}) < \infty$. Then the agnostic transductive regret for $\mathcal{H}$ is*

$$\Omega\left(\sqrt{\mathsf{VC}(\mathcal{H}) \cdot n}\right) \leq R(\mathcal{H}, n) \leq O\left(\sqrt{\mathsf{VC}(\mathcal{H}) \cdot n \cdot \log n}\right).$$

## 1.4 Related Works

The general idea of bounding the number of mistakes by learning algorithms in sequential prediction problems was introduced in the seminal work of Littlestone [Lit87]. That work introduced the *online* learning model, where the sequence of examples is revealed to the learner one example at a time. After each example $x$ is revealed, the learner makes a prediction, after which the true target label $y$ is revealed. The constraint, which makes learning even plausible, is that this sequence of $(x, y)$ pairs should maintain the property that there is an (unknown) target concept in a given concept class $\mathcal{H}$ which is correct on the entire sequence. Littlestone [Lit87] also identified the optimal predictor for this problem (called the *SOA*, for *Standard Optimal Algorithm*, and a general complexity measure which is precisely equal to the optimal bound on the number of mistakes: a quantity now referred to as the *Littlestone dimension*.

Later works discussed variations on this framework. In particular, as mentioned, the transductive model discussed in the present work was introduced in the work of [BKM97]. The idea (and terminology) of transductive learning was introduced by [VC74, Vap82, Kuh99], to capture scenarios

where learning may be easier due to knowing in advance which examples the learner will be tested on. [VC74, Vap82, Kuh99] study transductive learning in a model closer in spirit to the PAC framework, where some uniform random subset of examples have their labels revealed to the learner and it is tasked with predicting the labels of the remaining examples. In contrast, [BKM97] study transductive learning in a sequential prediction setting, analogous to the online learning framework of Littlestone. In this case, the sequence of examples $x$ is revealed to the learner all at once, and only the target labels (the $y$'s) are revealed in an online fashion, with the label of each example revealed just after its prediction for that example in the given sequential order. Since a mistake bound in this setting is still required to hold for *any* sequence, for the purpose of analysis we may think of the sequence of $x$'s as being a *worst case* set of examples and ordering thereof, for a given learning algorithm. [BKM97] compare and contrast the optimal mistake bound for this setting to that of the original online model of [Lit87]. Denoting by $d$ the Littlestone dimension of the concept class, it is clear that the optimal mistake bound would be no larger than $d$. However, they also argue that the optimal mistake bound in the transductive model is never smaller than $\Omega(\sqrt{\log(d)})$ (as mentioned, we improve this to $\log(d)$ in the present work). They further exhibit a family of concept classes of variable $d$ for which the transductive mistake bound is strictly smaller by a factor of $\frac{3}{2}$. They additionally provide a general equivalent description of the optimal transductive mistake bound in terms of the maximum possible rank among a certain family of trees, each representing the game tree for the sequential game on a given sequence of examples $x$.

In addition to these two models of sequential prediction, the online learning framework has also been explored in other variations, including exploring the optimal mistake bound under a *best-case* order of the data sequence $x$, or even a *self-directed* adaptive order in which the learning algorithm selects the next point for prediction from the remaining $x$'s from the given sequence on each round. [BKM97, BEK95, GS94, BE98, Kuh99].

Unlike the online learning model of Littlestone, the transductive model additionally allows for nontrivial mistake bounds in terms of the sequence *length* $n$ (the online model generally has $\min\{d, n\}$ as the optimal mistake bound). In this case, it follows immediately from the Sauer–Shelah–Perles lemma and a Halving technique that the optimal transductive mistake bound is no larger than $O(v \log(n))$ [KK05], where $v$ is the VC dimension of the concept class [VC71, VC74].

## 2  Preliminaries

**Notation 2.1.** *Let $\mathcal{X}$ be a set and $n, k \in \mathbb{N}$. For a sequence $x = (x_1, \ldots, x_n) \in \mathcal{X}^n$, we write $x_{\leq k}$ to denote the subsequence $(x_1, \ldots, x_k)$. If $k \leq 0$ then $x_{\leq k}$ denotes the empty sequence, $\mathcal{X}^0$.*

**Definition 2.2.** *Let $k \in \mathbb{N}$, let $\mathcal{X}$ and $\mathcal{Y}$ be sets, and let $\mathcal{H} \subseteq \mathcal{Y}^{\mathcal{X}}$. A sequence $(x_1, y_1), \ldots, (x_k, y_k) \in (\mathcal{X} \times \mathcal{Y})^k$ is $\underline{\text{realizable by } \mathcal{H}}$, or $\underline{\mathcal{H}\text{-realizable}}$, if $\exists h \in \mathcal{H} \ \forall i \in [k] : \ h(x_i) = y_i$.*

**Definition 2.3.** *Let $\mathcal{X}$ be a set, let $\mathcal{H} \subseteq \{0, 1\}^{\mathcal{X}}$, let $d \in \mathbb{N}$, and let $X = \{x_1, \ldots, x_d\} \subseteq \mathcal{X}$. We say that $\underline{\mathcal{H} \text{ shatters } X}$ if for every $y \in \{0, 1\}^d$ there exists $h \in \mathcal{H}$ such that for all $i \in [d]$, $h(x_i) = y_i$. The $\underline{\text{Vapnik–Chervonenkis (VC) dimension}}$ of $\mathcal{H}$ is $\mathsf{VC}(\mathcal{H}) = \sup\{|X| : \ X \subseteq \mathcal{X} \text{ finite} \wedge \mathcal{H} \text{ shatters } X\}$.*

**Definition 2.4** ([Lit87]). *Let $\mathcal{X}$ be a set and let $d \in \mathbb{N}$. A $\underline{\text{Littlestone tree of depth } d}$ with domain $\mathcal{X}$ is a set*

$$T = \left\{ x_u \in \mathcal{X} : \ u \in \bigcup_{s=0}^{d} \{0, 1\}^s \right\}. \tag{2}$$

*Let $\mathcal{H} \subseteq \{0, 1\}^{\mathcal{X}}$. We say that $\underline{\mathcal{H} \text{ shatters a tree } T}$ as in Eq. (2) if for every $u \in \{0, 1\}^{d+1}$ there exists $h_u \in \mathcal{H}$ such that*

$$\forall i \in [d+1] : \ h(x_{u_{\leq i-1}}) = u_i.$$

*The $\underline{\text{Littlestone dimension}}$ of $\mathcal{H}$, denoted $\mathsf{LD}(\mathcal{H})$, is the supremum over all $d \in \mathbb{N}$ such that there exists a Littlestone tree of depth $d$ with domain $\mathcal{X}$ that is shattered by $\mathcal{H}$.*

**Theorem 2.5** ([Lit87]). *Let $\mathcal{X}$ be a set and let $\mathcal{H} \subseteq \{0, 1\}^{\mathcal{X}}$ such that $d = \mathsf{LD}(\mathcal{H}) < \infty$. Then there exists a strategy for the learner that guarantees that the learner will make at most $d$ mistakes in the standard (non-transductive) online learning setting, regardless of the adversary's strategy and of number of instances to be labeled.*

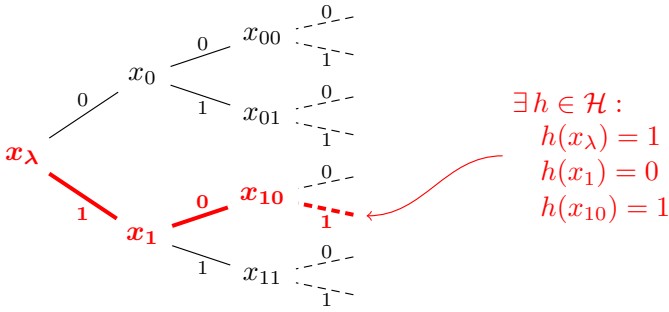

Figure 1: A shattered Littlestone tree of depth 2. The empty sequence is denoted by $\lambda$.

(Source: [BHM$^+$21])

**Theorem 2.6** (Sauer–Shelah–Perles; [She72, Sau72]). *Let $n, d \in \mathbb{N}$, let $\mathcal{X}$ be a set of cardinality $n$, and let $\mathcal{H} \subseteq \{0, 1\}^{\mathcal{X}}$ such that $\mathsf{VC}(\mathcal{H}) = d$. Then $|\mathcal{H}| \leq \sum_{i=0}^{n} \binom{n}{i} \leq \left(\frac{en}{d}\right)^d$.*

# 3 Quantitative Bounds

## 3.1 Littlestone Dimension: A Tighter Lower Bound

The Littlestone dimension is an upper bound on the number of mistakes, namely

$$\forall n \in \mathbb{N}: \ M(\mathcal{H}, n) \leq \mathsf{LD}(\mathcal{H}) \tag{3}$$

for any class $\mathcal{H}$. This holds because $\mathsf{LD}(\mathcal{H})$ is an upper bound on the number of mistakes for standard (non-transductive) online learning [Lit87], and the adversary in the transductive setting is strictly weaker.

The Littlestone dimension also supplies a lower bound. We give a quadratic improvement on the previous lower bound of [BKM97], as follows.

**Theorem 3.1.** *Let $\mathcal{X}$ be a set, let $\mathcal{H} \subseteq \{0, 1\}^{\mathcal{X}}$ such that $d = \mathsf{LD}(\mathcal{H}) < \infty$, and let $n \in \mathbb{N}$. Then*

$$M(\mathcal{H}, n) \geq \min\left\{\lfloor \log(d)/2 \rfloor, \lfloor \log\log(n)/2 \rfloor\right\}.$$

*Proof idea for Theorem 3.1.* Let $T$ be a Littlestone tree of depth $d$ that is shattered by $\mathcal{H}$, and let $\mathcal{H}_1 \subseteq \mathcal{H}$ be a collection of $2^{d+1}$ functions that witness the shattering. The adversary selects the sequence consisting of the nodes of $T$ in breadth-first order. For each time step $t \in [n]$, let $\mathcal{H}_t$ denote the version space, i.e., the subset of $\mathcal{H}_1$ that is consistent with all previously-assigned labels. The adversary's adaptive labeling strategy at time $t$ is as follows. If $\mathcal{H}_t$ is very unbalanced, meaning that a large majority of functions in $\mathcal{H}_t$ assign the same value to $x_t$, then the adversary chooses $y_t$ to be that value. Otherwise, if $\mathcal{H}_t$ is fairly balanced, the adversary forces a mistake (it can do so without violating $\mathcal{H}$-realizability). The pivotal observation is that: (1) under this strategy, the version space decreases in cardinality significantly more during steps where the adversary forces a mistake compared to steps where it did not force a mistake; (2) let $x_t$ be the $t$-th node in the breadth-first order. It has distance $\ell = \lfloor \log(t) \rfloor$ from the root of $T$. Because $T$ is a binary tree, the subtree $T'$ of $T$ rooted at $x_t$ is a tree of depth $d - \ell$. In particular, seeing as $\mathcal{H}_t$ contains only functions necessary for shattering $T'$, $|\mathcal{H}_t| \leq 2^{d-\ell+1}$, so $\mathcal{H}_t$ must decrease not too slowly with $t$. Combining (1) and (2) yields that the adversary must be able to force a mistake not too rarely. A careful quantitative analysis shows that the number of mistakes the adversary can force is at least logarithmic in $d$. $\qquad\square$

The full proof of Theorem 3.1 appears in Appendix A.

## 3.2 Threshold Dimension

We also show some bounds on the number of mistakes in terms of the threshold dimension.

**Definition 3.2.** *Let $\mathcal{X}$ be a set, let $X = \{x_1, \ldots, x_k\} \subseteq \mathcal{X}$, and let $\mathcal{H} \subseteq \{0, 1\}^{\mathcal{X}}$. We say that $X$ is threshold-shattered by $\mathcal{H}$ if there exist $h_1, \ldots, h_k \in \mathcal{H}$ such that $h_i(x_j) = \mathbb{1}(j \leq i)$ for all $i, j \in [k]$.*

*The underline{threshold dimension of} $\mathcal{H}$, denoted* $\mathsf{TD}(\mathcal{H})$*, is the supremum of the set of integers $k$ for which there exists a threshold-shattered set of cardinality $k$.*

The following connection between the threshold and Littlestone dimensions is well-known.

**Theorem 3.3** ([She90, Hod97]). *Let $\mathcal{X}$ be a set, let $\mathcal{H} \subseteq \{0,1\}^{\mathcal{X}}$, and let $d \in \mathbb{N}$. Then:*

  1. *If $\mathsf{LD}(\mathcal{H}) \geq d$ then $\mathsf{TD}(\mathcal{H}) \geq \lfloor \log d \rfloor$.*
  2. *If $\mathsf{TD}(\mathcal{H}) \geq d$ then $\mathsf{LD}(\mathcal{H}) \geq \lfloor \log d \rfloor$.*

Item 1 in Theorem 3.3 and Eq. (3) imply that

$$\forall n \in \mathbb{N}: \ M(\mathcal{H}, n) \leq 2^{\mathsf{TD}(\mathcal{H})}$$

for any class $\mathcal{H}$. Similarly, Item 2 in Theorem 3.3 and Theorem 3.1 imply a mistake lower bound of $\Omega(\log \log(\mathsf{TD}(\mathcal{H})))$. However, one can do exponentially better than that, as follows.

**Claim 3.4.** *Let $\mathcal{X}$ be a set, let $\mathcal{H} \subseteq \{0,1\}^{\mathcal{X}}$ such that $d = \mathsf{TD}(\mathcal{H}) < \infty$, and let $n \in \mathbb{N}$. Then*

$$M(\mathcal{H}, n) \geq \min\left\{ \lfloor \log(d) \rfloor, \lfloor \log(n) \rfloor \right\}.$$

One of the ideas used in this proof appeared in an example called $\sigma_{\text{worst}}$ in Section 4.1 of [BKM97].

$$
\begin{array}{ll}
q_1: & x_{\frac{N}{2}} \\[1em]
q_2: & x_{\frac{N}{4}} \qquad\qquad x_{\frac{3N}{4}} \\[1em]
q_3: & x_{\frac{N}{8}} \quad x_{\frac{3N}{8}} \quad x_{\frac{5N}{8}} \quad x_{\frac{7N}{8}} \\[1em]
& \vdots \qquad\qquad \vdots
\end{array}
$$

Figure 2: Construction of the sequence $q$ in the proof of Claim 3.4. $q$ is a breadth-first enumeration of the depicted binary tree.

*Proof of Claim 3.4.* Let $k = \min\left\{\lfloor \log(d) \rfloor, \lfloor \log(n) \rfloor\right\}$ and let $N = 2^k$. Let $X = \{x_1, \ldots, x_{N-1}\} \subseteq \mathcal{X}$ be a set that is threshold-shattered by functions $h_1, \ldots, h_{N-1} \in \mathcal{H}$ and $h_i(x_j) = \mathbb{1}(j \leq i)$ for all $i, j \in [N-1]$. The strategy for the adversary is to present $X$ in dyadic order, namely

$$x_{\frac{N}{2}}, x_{\frac{N}{4}}, x_{\frac{3N}{4}}, x_{\frac{N}{8}}, x_{\frac{3N}{8}}, x_{\frac{5N}{8}}, x_{\frac{7N}{8}}, \ldots, x_{\frac{(2^k-1)N}{2^k}}.$$

More explicitly, the adversary chooses the sequence $q = q_1 \circ q_2 \circ \cdots \circ q_k$, where '$\circ$' denotes sequence concatenation and

$$q_i = \left( x_{\frac{1}{2^i}N}, x_{\frac{3}{2^i}N}, x_{\frac{5}{2^i}N}, x_{\frac{7}{2^i}N}, \ldots, x_{\frac{(2^i-1)}{2^i}N} \right)$$

for all $i \in [k]$. See Figure 2.

We prove by induction that for each $i \in [k]$, all labels chosen by the adversary for the subsequences prior to $q_i$ are $\mathcal{H}$-realizable, and additionally there exists an instance in subsequence $q_i$ on which the adversary can force a mistake regardless of the learners predictions. The base case is that the adversary can always force a mistake on the first instance, $q_1$, by choosing the label opposite to the learner's prediction (both labels 0 and 1 are $\mathcal{H}$-realizable for this instance). Subsequently, for any $i > 1$, note that by the induction hypothesis, the labels chosen by the adversary for all instances in the previous subsequences are $\mathcal{H}$-realizable. In particular there exists an index $a \in [N]$ such that instance $x_a$ has already been labeled, and all the labels chosen so far are consistent with $h_a$. Let $b$ be the minimal integer such that $b > a$ and $x_b$ has also been labeled. Then $x_a$ and all labeled instances with smaller indices received label 1, while $x_b$ and all labeled instances with larger indices received label 0. Because the sequence is dyadic, subsequence $q_i$ contains an element $x_m$ such that $a < m < b$. The adversary can force a mistake on $x_m$, because $h_a$ and $h_m$ agree on all previously labeled instances, but disagree on $x_m$. $\qquad\square$

Claim 3.4 is used in the proof of the trichotomy (Theorem 4.1, below).

Finally, we note that for every $d \in \mathbb{N}$ there exists a hypothesis class $\mathcal{H}$ such that $d = \mathsf{TD}(\mathcal{H})$ and
$$\forall n \in \mathbb{N} : \ M(\mathcal{H}, n) = \min\{d, n\}.$$
Indeed, take $\mathcal{X} = [d]$ and $\mathcal{H} = \{0, 1\}^{\mathcal{X}}$. The upper bound holds because $|\mathcal{X}| = d$, and the lower bound holds by Item 2 in Theorem 4.1, because $\mathsf{VC}(\mathcal{H}) = d$.

## 4 Trichotomy

**Theorem 4.1.** *Let $\mathcal{X}$ be a set, let $\mathcal{H} \subseteq \{0, 1\}^{\mathcal{X}}$, and let $n \in \mathbb{N}$ such that $n \leq |\mathcal{X}|$.*

1. *If $\mathsf{VC}(\mathcal{H}) = \infty$ then $M(\mathcal{H}, n) = n$.*

2. *Otherwise, if $\mathsf{VC}(\mathcal{H}) = d < \infty$ and $\mathsf{LD}(\mathcal{H}) = \infty$ then*
$$\max\{\min\{d, n\}, \lfloor \log(n) \rfloor\} \leq M(\mathcal{H}, n) \leq O(d \log(n/d)). \tag{4}$$
   *Furthermore, each of the bounds in Eq. (4) is tight for some classes. The $\Omega(\cdot)$ and $O(\cdot)$ notations hide universal constants that do not depend on $\mathcal{X}$ or $\mathcal{H}$.*

3. *Otherwise, there exists an integer $C(\mathcal{H}) \leq \mathsf{LD}(\mathcal{H})$ (that depends on $\mathcal{X}$ and $\mathcal{H}$ but does not depend on $n$) such that $M(\mathcal{H}, n) \leq C(\mathcal{H})$.*

*Proof of Theorem 4.1.* For Item 1, assume $\mathsf{VC}(\mathcal{H}) = \infty$. Then there exists a set $X = \{x_1, \ldots, x_n\} \subseteq \mathcal{X}$ of cardinality $n$ that is shattered by $\mathcal{H}$. The adversary can force the learner to make $n$ mistakes by selecting the sequence $(x_1, \ldots, x_n)$, and then selecting labels $y_t = 1 - \hat{y}_t$ for all $t \in [n]$. This choice of labels is $\mathcal{H}$-realizable because $X$ is a shattered set.

To obtain the upper bound in Item 2 the learner can use the *halving algorithm*, as follows. Let $x = (x_1, \ldots, x_n)$ be the sequence chosen by the adversary, and let $\mathcal{H}|_x$ denote the collection of functions from elements of $x$ to $\{0, 1\}$ that are restrictions of functions in $\mathcal{H}$. For each $t \in \{0, \ldots, n\}$, let
$$\mathcal{H}_t = \big\{ f \in \mathcal{H}|_x : \ (\forall i \in [t] : \ f(x_i) = y_i) \big\}$$
be a set called the *version space* at time $t$. At each step $t \in [n]$, the learner makes prediction
$$\hat{y}_t = \mathrm{argmax}_{b \in \{0,1\}} \big| \{ f \in \mathcal{H}_{t-1} : \ f(x_t) = b \} \big|.$$
In words, the learner chooses $\hat{y}_t$ according to a majority vote among the functions in version space $\mathcal{H}_{t-1}$, and then any function whose vote was incorrect is excluded from the next version space, $\mathcal{H}_t$. This implies that for any $t \in [n]$, if the learner made a mistake at time $t$ then
$$|\mathcal{H}_t| \leq \frac{1}{2} \cdot |\mathcal{H}_{t-1}|. \tag{5}$$
Let $M = M(\mathcal{H}, n)$. The adversary selects $\mathcal{H}$-realizable labels, so $\mathcal{H}_n$ cannot be empty. Hence, applying Eq. (5) recursively yields
$$1 \leq |\mathcal{H}_n| \leq 2^{-M} \cdot |\mathcal{H}_0| \leq 2^{-M} \cdot O\big((n/d)^d\big),$$
where the last inequality follows from $\mathsf{VC}(\mathcal{H}_0) \leq \mathsf{VC}(\mathcal{H}) = d$ and the Sauer–Shelah–Perles lemma (Theorem 2.6). Hence $M = O(d \log(n/d))$, as desired.

For the $\min\{d, n\}$ lower bound in Item 2, if $n \leq d$ then the adversary can force $n$ mistakes by the same argument as in Item 1. For the logarithmic lower bound in Item 2, the assumption that $\mathsf{LD}(\mathcal{H}) = \infty$ and Theorem 3.3 imply that $\mathsf{TD}(\mathcal{H}) = \infty$, and in particular $\mathsf{TD}(\mathcal{H}) \geq n$, and this implies the bound by Claim 3.4.

For Item 3, the assumption $\mathsf{LD}(\mathcal{H}) = k < \infty$ and Theorem 2.5 imply that for any $n$, the learner will make at most $k$ mistakes. This is because the adversary in the transductive setting is strictly weaker than the adversary in the standard online setting. So there exists some $C(\mathcal{H}) \in \{0, \ldots, k\}$ as desired. $\qquad \square$

**Remark 4.2.** *One can use Theorem 3.1 to obtain a lower bounds for the case of Item 2 in Theorem 4.1. However, that yields a lower bound of $\Omega(\log \log(n))$, which is exponentially weaker than the bound we show.*

# 5 Multiclass Setting

The trichotomy of Theorem 4.1 can be generalized to the multiclass setting, in which the label set $\mathcal{Y}$ contains more than two labels. In this setting, the VC dimension is replaced by the Natarajan dimension [Nat89], denoted ND, and the Littlestone dimension is generalized in the natural way. The result holds for *finite* sets $\mathcal{Y}$.

**Theorem 5.1** (Informal Version of Theorem B.3). *Let $\mathcal{X}$ be a set, let $\mathcal{Y}$ be a* finite *set, and let $\mathcal{H} \subseteq \mathcal{Y}^{\mathcal{X}}$. Then $\mathcal{H}$ satisfies precisely one of the following:*

1. *$M(\mathcal{H}, n) = n$. This happens if $\mathsf{ND}(\mathcal{H}) = \infty$.*

2. *$M(\mathcal{H}, n) = \Theta(\log(n))$. This happens if $\mathsf{ND}(\mathcal{H}) < \infty$ and $\mathsf{LD}(\mathcal{H}) = \infty$.*

3. *$M(\mathcal{H}, n) = O(1)$. This happens if $\mathsf{LD}(\mathcal{H}) < \infty$.*

The proof of Theorem 5.1 appears in Appendix B, along with the necessary definitions. The main innovation in the proof involves the use of the multiclass threshold bounds developed in Appendix D, which in turn rely on a basic result from Ramsey theory, stated Appendix C.

## 5.1 The Case of an Infinite Label Set

It is interesting to observe that the analogy between the binary classification and multiclass classification settings breaks down when the label set $\mathcal{Y}$ is not finite.

**Example 5.2.** *There exists a class $\mathcal{G} \subseteq \mathcal{Y}^{\mathcal{X}}$ such that $\mathcal{Y}$ is countable, $\mathsf{LD}(\mathcal{G})$ is infinite, but the class is learnable with a mistake bound of $M(\mathcal{G}, n) = 1$. To see this, let $\mathcal{X}$ be countable, and let $\mathcal{H} \subseteq \{0,1\}^{\mathcal{X}}$ be a class with $\mathsf{LD}(\mathcal{H}) = \infty$. For each $i \in \mathbb{N}$, let $T_i$ be a Littlestone tree of depth $i$ that is shattered by $\mathcal{H}$, and let $\{h_1^i, \ldots, h_{2^{i+1}}^i\} \subseteq \mathcal{H}$ be a subset that witnesses the shattering. Let $\mathcal{G} = \{g_j^i : i \in \mathbb{N} \wedge j \in [2^{i+1}]\}$ be a set of functions such that $g_j^i(x) = (h_j^i(x), i, j)$ for all $i, j$. Let $\mathcal{Y} = \{0,1\} \times \mathbb{N} \times \mathbb{N}$. Observe that $\mathcal{G} \subseteq \mathcal{Y}^{\mathcal{X}}$ is a countable set of functions with a countable set of labels. Furthermore, $\mathsf{LD}(\mathcal{G}) = \infty$ because $\mathcal{G}$ shatters a sequence of suitable Littlestone trees corresponding to $T_1, T_2, \ldots$. However, $\mathcal{G}$ can be learned with mistake bound 1, because a single example of the form $\left(x, (h_j^i(x), i, j)\right)$ reveals the correct labeling function $h_j^i$.*

Recent work by [BCD+22] has shown that multiclass PAC learning with infinite $\mathcal{Y}$ is not characterized by the Natarajan dimension, and that instead it is characterized by the DS dimension (introduced by [DS14]). It is therefore natural to ask whether the DS dimension might also characterize multiclass transductive online learning with infinite $\mathcal{Y}$. We show that the answer to that question is negative.

Recall the definition of the DS dimension.

**Definition 5.3.** *Let $d \in \mathbb{N}$, let $\mathcal{X}$ and $\mathcal{Y}$ be sets, and let $\mathcal{H} \subseteq \mathcal{Y}^{\mathcal{X}}$. For an index $i \in [d]$ and vectors $y = (y_1, \ldots, y_d) \in \mathcal{Y}^d$, $y' = (y_1', \ldots, y_d') \in \mathcal{Y}^d$, we say that y and y' are i-neighbors, denoted $y \sim_i y'$, if $\{j \in [d] : y_j \neq y_j'\} = \{i\}$. We say that $\mathcal{C} \subseteq \mathcal{Y}^d$ is a d-pseudocube if $\mathcal{C}$ is non-empty and finite, and*

$$\forall y \in \mathcal{C} \, \forall i \in [d] \, \exists y' \in \mathcal{C} : \, y \sim_i y'.$$

*For a vector $x = (x_1, \ldots, x_d) \in \mathcal{X}^d$, we say that $\mathcal{H}$ DS-shatters x if the set*

$$\mathcal{H}|_x := \left\{\left(h(x_1), \ldots, h(x_d)\right) : \, h \in \mathcal{H}\right\} \subseteq \mathcal{Y}^d$$

*contains a d-pseudocube.*

*Finally, the Daniely–Shalev-Shwartz (DS) dimension of $\mathcal{H}$ is*

$$\mathsf{DS}(\mathcal{H}) = \sup \left\{d \in \mathbb{N} : \, \left(\exists x \in \mathcal{X}^d : \, \mathcal{H} \text{ DS-shatterd } x\right)\right\}.$$

See [BCD+22] for figures and further discussion of the DS dimension.

The following claim shows that the DS dimension does not characterize transductive online learning, even when $\mathcal{Y}$ is finite.

**Claim 5.4.** *For every $n \in \mathbb{N}$, there exists a hypothesis class $\mathcal{H}_n$ such that $\mathsf{DS}(\mathcal{H}_n) = 1$ but the adversary in transductive online learning can force at least $M(\mathcal{H}_n, n) = n$ mistakes.*

*Proof.* Fix $n \in \mathbb{N}$ and let $\mathcal{X} = \{0, 1, 2, \ldots, n\}$. Consider a complete binary tree $T$ of depth $n$ such that for each $x \in \mathcal{X}$, all the nodes at depth $x$ (at distance $x$ from the root) are labeled by $x$, and each edge in $T$ is labeled by a distinct label. Let $\mathcal{H}$ be a minimal hypothesis class that shatters $T$, namely, $\mathcal{H}$ shatters $T$ and there does not exist a strict subset of $\mathcal{H}$ that shatters $T$.

Observe that $M(\mathcal{H}_n, n) = n$, because the adversary can present the sequence $0, 1, 2, \ldots, n - 1$ and force a mistake at each time step. To see that $\mathsf{DS}(\mathcal{H}_n) = 1$, assume for contradiction that there exists a vector $x = (x_1, x_2) \in \mathcal{X}^2$ that is DS-shattered by $\mathcal{H}_n$, namely, there exists a 2-pseudocube $\mathcal{C} \subseteq \mathcal{H}|_x$. Note that $x_1 \neq x_2$, and without loss of generality $x_1 < x_2$ ($\mathcal{H}$ DS-shatters $(x_1, x_2)$ if and only if it DS-shatters $(x_2, x_1)$).

Fix $y \in \mathcal{C}$. So $y = (h(x_1), h(x_2))$ for some $h \in \mathcal{H}$. Because $\mathcal{C}$ is a 2-pseudocube, there exists $y' \in \mathcal{C}$ that is a 1-neighbor of $y$. Namely, there exists $g \in \mathcal{H}$ such that $y' = (g(x_1), g(x_2)) \in \mathcal{C}$, $y_1' \neq y_1$ and $y_2' = y_2$. However, because each edge in $T$ has a distinct label, and $\mathcal{H}$ is minimal, it follows that for any $x \in \mathcal{X}$,

$$g(x) = h(x) \implies \big(\forall x' \in \{0, 1, \ldots, x\} : g(x') = h(x')\big).$$

In particular, $g(x_2) = y_2' = y_2 = h(x_2)$ implies $y_1' = g(x_1) = h(x_1) = y_1$ which is a contradiction to the choice of $y'$. □

## 6 Agnostic Setting

The *agnostic* transductive online learning setting is defined analogously to the *realizable* (non-agnostic) transductive online learning setting described in Section 1.1. An early work by [Cov65] observed that it is not possible for a learner to achieve vanishing regret in an agnostic online setting with complete information. Therefore, we consider a game with incomplete information, as follows.

First, the adversary selects an arbitrary sequence of instances, $x_1, \ldots, x_n \in \mathcal{X}$, and reveals the sequence to the learner. Then, for each $t = 1, \ldots, n$:

1. The adversary selects a label $y_t \in \mathcal{Y}$.

2. The learner selects a prediction $\hat{y}_t \in \mathcal{Y}$ and reveals it to the adversary.

3. The adversary reveals $y_t$ to the learner.

At each time step $t \in [n]$, the adversary may select any $y_t \in \mathcal{Y}$, without restrictions.[2] The learner, which is typically randomized, has the objective of minimizing the *regret*, namely

$$R(A, \mathcal{H}, x, y) = \mathbb{E}[|\{t \in [n] : \hat{y}_t \neq y_t\}|] - \min_{h \in \mathcal{H}} |\{t \in [n] : h(x_t) \neq y_t\}|,$$

where the expectation is over the learner's randomness. In words, the regret is the expected excess number of mistakes the learner makes when it plays strategy $A$ and the adversary chooses the sequence $x \in \mathcal{X}^n$ and labels $y \in \mathcal{Y}^n$, as compared to the number of mistakes made by the best fixed hypothesis $h \in \mathcal{H}$. We are interested in understanding the value of this game, namely

$$R(\mathcal{H}, n) = \inf_{A \in \mathcal{A}} \sup_{x \in \mathcal{X}^n} \sup_{y \in \mathcal{Y}^n} R(A, \mathcal{H}, x, y),$$

where $\mathcal{A}$ is the set of all learner strategies. We show the following result.

**Theorem 6.1.** *Let $\mathcal{X}$ be a set, let $\mathcal{H} \subseteq \{0, 1\}^{\mathcal{X}}$, and let $n \in \mathbb{N}$ such that $n \leq |\mathcal{X}|$. Assume $0 < \mathsf{VC}(\mathcal{H}) < \infty$. Then the agnostic transductive regret for $\mathcal{H}$ on sequences of length $n$ is*

$$\Omega\left(\sqrt{\mathsf{VC}(\mathcal{H}) \cdot n}\right) \leq R(\mathcal{H}, n) \leq O\left(\sqrt{\mathsf{VC}(\mathcal{H}) \cdot n \cdot \log\left(n/\mathsf{VC}(\mathcal{H})\right)}\right).$$

The upper bound in Theorem 6.1 follows directly from the the Sauer–Shelah–Perles lemma (Theorem 2.6), together with the following well-known bound on the regret of the *Multiplicative Weights* algorithm (see, e.g., Theorem 21.10 in [SB14]).

**Theorem 6.2.** *Let $\mathcal{X}$ be a set and let $\mathcal{H} \subseteq \{0, 1\}^{\mathcal{X}}$ be finite. There exists an algorithm for the standard (non-transductive) agnostic online learning setting that satisfies*

$$R_{\mathsf{online}}(\mathcal{H}, n) \leq \sqrt{2 \log\left(|\mathcal{H}|\right)}.$$

---

[2]Hence the name 'agnostic', implying that we make no assumptions concerning the choice of labels.

Theorem 6.2 implies the upper bound of Theorem 6.1, because the adversary in the transductive agnostic setting is weaker than the adversary in the standard agnostic setting.

We prove the lower bound of Theorem 6.1 using an anti-concentration technique from Lemma 14 of [BPS09]. The proof appears in Appendix E.

**Remark 6.3.** *Additionally:*

1. *If* $\mathsf{VC}(\mathcal{H}) = 0$ *(i.e., classes with a single function) then the regret is* $0$.

2. *If* $\mathsf{VC}(\mathcal{H}) < \infty$ *and* $\mathsf{LD}(\mathcal{H}) < \infty$ *then the regret is* $R(\mathcal{H}, n) = O\left(\sqrt{\mathsf{LD}(\mathcal{H}) \cdot n}\right)$, *by* [ABD+21] *(as mentioned above). Namely, in some cases the* $\log(n)$ *factor in Theorem* 6.1 *can be removed.*

3. *If* $\mathsf{VC}(\mathcal{H}) = \infty$ *then the regret is* $\Omega(n)$.

# 7 Future Work

Some remaining open problems include:

1. Showing a sharper bound for the $\Theta(1)$ case in the trichotomy (Theorem 4.1). Currently, there is an exponential gap between the best known upper and lower bounds for Littlestone classes.

2. Showing sharper bounds for the $\Theta(\log n)$ cases in the trichotomy (Theorem 4.1) and multiclass trichotomy (Theorem B.3).

3. Showing a sharper bound for the agnostic case (Theorem 6.1).

4. Characterizing the numeber of mistakes in the multiclass setting with an infinite label set $\mathcal{Y}$ (complementing Theorem B.3).

# Acknowledgments

Zachary Chase contributed significantly to this paper. All errors are our own.

SM is a Robert J. Shillman Fellow; he acknowledges support by ISF grant 1225/20, by BSF grant 2018385, by an Azrieli Faculty Fellowship, by Israel PBC-VATAT, by the Technion Center for Machine Learning and Intelligent Systems (MLIS), and by the European Union (ERC, GENERAL-IZATION, 101039692). JS was supported by DARPA (Defense Advanced Research Projects Agency) contract # HR001120C0015 and the Simons Collaboration on The Theory of Algorithmic Fairness. Views and opinions expressed are those of the author(s) only and do not necessarily reflect those of the European Union, the European Research Council Executive Agency, DARPA, or the Simons Foundation. The European Union, the granting authority, DARPA, and the Simons Foundation cannot be held responsible for them.

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

# Appendices

## A    Proof of Lower Bound for Littlestone Classes

*Proof of Theorem 3.1.* Let $T$ be a Littlestone tree of depth $d$ that is shattered by $\mathcal{H}$, and let $\mathcal{H}_1 \subseteq \mathcal{H}$ be a collection of $2^{d+1}$ functions that witness the shattering. $T$ contains $n_T = 2^{d+1} - 1$ nodes. The adversary selects the sequence

$$x_1, x_2, \ldots, x_n$$

consisting of the first $n$ nodes of $T$ in breadth-first order (if $n > n_T$, then the adversary chooses the suffix $x_{n_T+1}, \ldots, x_n$ arbitrarily). For each time step $t \in [n]$, let $\mathcal{H}_t$ denote the version space, i.e., the subset of $\mathcal{H}_1$ that is consistent with all previously-assigned labels. Namely, for any $t > 1$,

$$\mathcal{H}_t = \{h \in \mathcal{H}_1 : (\forall s \in [t-1] : h(x_s) = y_s)\}.$$

Similarly, for each $b \in \{0, 1\}$, let $\mathcal{H}_{t,b} = \{h \in \mathcal{H}_t : h(x_t) = b\}$.

---

**send** $x_1, \ldots, x_n$ to learner

$k \leftarrow 1$

**for** $t \leftarrow 1, 2, \ldots, n$:

    $m_k \leftarrow 2^{2^{2k}}$

    $\varepsilon_t \leftarrow 1/m_k$

    $r_t \leftarrow |\mathcal{H}_{t,1}|/|\mathcal{H}_t|$

    **receive** $\hat{y}_t$ from learner

    $y_t \leftarrow \begin{cases} 1 - \hat{y}_t & r_t \in [\varepsilon_t, 1 - \varepsilon_t] \\ 0 & r_t \in [0, \varepsilon_t) \\ 1 & r_t \in (1 - \varepsilon_t, 1] \end{cases}$

    **send** $y_t$ to learner

    **if** $r_t \in [\varepsilon_t, 1 - \varepsilon_t]$:

        $k \leftarrow k + 1$

Algorithm 1: An adversary that forces $\Omega(\log(\mathsf{LD}(\mathcal{H})))$ mistakes.

The adversary operates according to Algorithm 1. Conceptually, at each time step $t \in [n]$, if $\mathcal{H}_t$ is very unbalanced, meaning that a large majority of the functions in $\mathcal{H}_t$ assign the same value to $x_t$, then the adversary chooses $y_t$ to be that value. Otherwise, if $\mathcal{H}_t$ is fairly balanced, the adversary forces a mistake. Note that if $\mathcal{H}_t$ is fairly balanced then the adversary can force a mistake without violating $\mathcal{H}$-realizability.

We now argue that using this strategy, the adversary forces $\Omega(\log(d))$ mistakes. Let $F = \{t_1, t_2, \ldots\} = \{t \in [n] : r_t \in [\varepsilon_t, 1 - \varepsilon_t]\}$ be the set of time steps where the adversary forces a mistake. Note that in the for-loop in Algorithm 1, the value of $k$ at the beginning of iteration $t_k$ is $k$ (e.g., at the beginning of iteration $t_3$, $k = 3$).

We argue by induction that for any $k \in \mathbb{N}$, if $m_k := 2^{2^{2k}} \leq n$ then:

1. $|F| \geq k$ and $t_k \leq m_k$; and

2. $|\mathcal{H}_{t_k}| \geq (1/m_k)^2 \cdot |\mathcal{H}_1|$.

The base case is immediate for $t_1 = 1 \in F$. For the induction step, assuming that Items 1 and 2 hold for some $k \in \mathbb{N}$ such that $m_{k+1} \leq n$, we show that they also hold for $k + 1$. For Item 1, assume for contradiction that $t \notin F$ for all $t$ such that $t_k < t \leq m_{k+1}$.

For each $t$, $t_k < t \le m_{k+1}$, the definition of $r_t$ and the adversary's labeling strategy imply that the label $y_t$ agrees with at least a $(1 - \varepsilon_t)$-majority of the functions in the version space $\mathcal{H}_t$. Hence,

$$
\begin{aligned}
\left|\mathcal{H}_{m_{k+1}}\right| &\ge \left|\mathcal{H}_{t_k}\right| \cdot \prod_{t=t_k+1}^{m_{k+1}} (1 - \varepsilon_t) \\
&= \left|\mathcal{H}_{t_k}\right| \cdot (1 - 1/m_{k+1})^{m_{k+1} - t_k} \\
&\ge \left|\mathcal{H}_{t_k}\right| \cdot (1 - 1/m_{k+1})^{m_{k+1}} \\
&\ge \left|\mathcal{H}_1\right| \cdot (1/m_k)^2 \cdot (1 - 1/m_{k+1})^{m_{k+1}} \qquad \text{(Induction hypothesis for Item 2)} \\
&\ge \left|\mathcal{H}_1\right| \cdot (1/m_k)^2 \cdot (1/4) \\
&= \left|\mathcal{H}_1\right| \cdot 2^{-2^{2k+1}-2}. \qquad\qquad\qquad\qquad\qquad\qquad\qquad\qquad\qquad\qquad (6)
\end{aligned}
$$

Observe that for every $t \in [n]$, if $x_t$ is a node with depth $\ell$ in $T$ (i.e., the shortest path from the root to $x_t$ contains $\ell$ edges), then there exists an 'active' node $x_\ell^*$ with the same depth $\ell$ in $T$ such that the version space $\mathcal{H}_t$ contains only functions from $\mathcal{H}_1$ that are consistent with the labels along the path from the root of $T$ to $x_\ell^*$. Namely, $\mathcal{H}_t$ is a subset of the $2^{d-\ell+1}$ functions in $\mathcal{H}_1$ that witness the shattering of the subtree $T_\ell$ of $T$ that is rooted at $x_\ell^*$. In particular, the depth (distance from the root) of node $x_{m_{k+1}}$ is $\log\left(2^{2^{2(k+1)}}\right) = 2^{2k+2}$, so

$$
\left|\mathcal{H}_{m_{k+1}}\right| \le 2^{d - 2^{2k+2} + 1} = 2^{d+1} \cdot 2^{-2^{2k+2}} = |\mathcal{H}_1| \cdot 2^{-2^{2k+2}}. \qquad (7)
$$

Combining Eqs. (6) and (7) yields $2^{-2^{2k+1}-2} \le 2^{-2^{2k+2}}$, which is a contradiction. This establishes Item 1. Item 2 follows by a similar calculation, which accounts for the fact that at time $t_{k+1}$ the adversary forces a mistake, and this reduces the version space by a factor of at most $\varepsilon_{t_{k+1}}$:

$$
\begin{aligned}
\left|\mathcal{H}_{t_{k+1}}\right| &\ge \left|\mathcal{H}_{t_k}\right| \cdot \left(\prod_{t=t_k+1}^{t_{k+1}-1} (1 - \varepsilon_t)\right) \cdot \varepsilon_{t_{k+1}} \\
&\ge \left|\mathcal{H}_{t_k}\right| \cdot (1 - 1/m_{k+1})^{m_{k+1}} \cdot (1/m_{k+1}) \\
&\ge \left|\mathcal{H}_{t_k}\right| \cdot (1/4) \cdot (1/m_{k+1}) \\
&\ge \left|\mathcal{H}_1\right| \cdot (1/m_k)^2 \cdot (1/4) \cdot (1/m_{k+1}) \qquad \text{(Induction hypothesis for Item 2)} \\
&= \left|\mathcal{H}_1\right| \cdot 2^{-2 \cdot 2^{2k}} \cdot (1/4) \cdot 2^{-2^{2k+2}} = |\mathcal{H}_1| \cdot 2^{-6 \cdot 2^{2k} - 2} \\
&\ge \left|\mathcal{H}_1\right| \cdot 2^{-8 \cdot 2^{2k}} = |\mathcal{H}_1| \cdot 2^{-2 \cdot 2^{2(k+1)}} = |\mathcal{H}_1| \cdot (1/m_{k+1})^2.
\end{aligned}
$$

This completes the induction.

To complete the proof, let $k^* = \min\left\{\lfloor \log(d)/2 \rfloor, \lfloor \log\log(n)/2 \rfloor\right\}$. Then $m_{k^*} \le 2^d < 2^{d+1} - 1 = n_T$, so $T$ contains at least $m_{k^*}$ nodes. Additionally, $m_{k^*} \le n$, so Item 1 implies that $|F| \ge k^*$, namely, the adversary can force at least $k^*$ mistakes, as desired. $\qquad\square$

## B  Multiclass Trichotomy

The following generalization of the Littlestone dimesion to the multiclass setting is due to [DSBS15].

**Definition B.1** (Multiclass Littlestone Dimension). *Let $\mathcal{X}$ and $\mathcal{Y}$ be sets and let $d \in \mathbb{N}$. A Littlestone tree of depth $d$ with domain $\mathcal{X}$ and label set $\mathcal{Y}$ is a set*

$$
T = \left\{(x_u, y_{u \circ 0}, y_{u \circ 1}) \in \mathcal{X} \times \mathcal{Y} \times \mathcal{Y} : \ u \in \bigcup_{s=0}^{d}\{0,1\}^s \ \wedge \ y_{u \circ 0} \ne y_{u \circ 1}\right\}, \qquad (8)
$$

*where '$\circ$' denotes string concatenation. Let $\mathcal{H} \subseteq \mathcal{Y}^\mathcal{X}$. We say that $\mathcal{H}$ shatters a tree $T$ as in Eq. (8) if for every $u \in \{0,1\}^{d+1}$ there exists $h_u \in \mathcal{H}$ such that*

$$
\forall i \in [d+1] : \ h(x_{u_{\le i-1}}) = y_{u_{\le i}}.
$$

*The Littlestone dimension of $\mathcal{H}$, denoted $\mathsf{LD}(\mathcal{H})$, is the supremum over all $d \in \mathbb{N}$ such that there exists a Littlestone tree of depth $d$ with domain $\mathcal{X}$ and label set $\mathcal{Y}$ that is shattered by $\mathcal{H}$.*

The Natarajan dimension is a popular generalization of the VC dimension to the multiclass setting.

**Definition B.2** ([Nat89])**.** *Let $\mathcal{X}$ and $\mathcal{Y}$ be sets, let $\mathcal{H} \subseteq \mathcal{Y}^{\mathcal{X}}$, let $d \in \mathbb{N}$, and let $X = \{x_1, \ldots, x_d\} \subseteq \mathcal{X}$. We say that $\mathcal{H}$ Natarajan-shatters $X$ if there exist $f_0, f_1 : X \to \mathcal{Y}$ such that:*

1. *$\forall x \in X : f_0(x) \neq f_1(x)$; and*

2. *$\forall A \subseteq X \ \exists h \in \mathcal{H} \ \forall x \in X : h(x) = f_{\mathbb{1}(x \in A)}(x)$.*

*The Natarajan dimension of $\mathcal{H}$ is $\mathsf{ND}(\mathcal{H}) = \sup \{|X| : X \subseteq \mathcal{X} \text{ finite } \wedge \ \mathcal{H} \text{ Natarajan-shatters } X\}$.*

We show the following generalization of Theorem 4.1 for the multiclass setting.

**Theorem B.3** (**Formal Version of Theorem 5.1**)**.** *Let $\mathcal{X}$ and $\mathcal{Y}$ be sets with $k = |\mathcal{Y}| < \infty$, let $\mathcal{H} \subseteq \mathcal{Y}^{\mathcal{X}}$, and let $n \in \mathbb{N}$ such that $n \leq |\mathcal{X}|$.*

1. *If $\mathsf{ND}(\mathcal{H}) = \infty$ then $M(\mathcal{H}, n) = n$.*

2. *Otherwise, if $\mathsf{ND}(\mathcal{H}) = d < \infty$ and $\mathsf{LD}(\mathcal{H}) = \infty$ then*
$$\max\{\min\{d, n\}, \lfloor \log(n) \rfloor\} \leq M(\mathcal{H}, n) \leq O(d \log(nk/d)). \tag{9}$$
   *The $\Omega(\cdot)$ and $O(\cdot)$ notations hide universal constants that do not depend on $\mathcal{X}$, $\mathcal{Y}$ or $\mathcal{H}$.*

3. *Otherwise, there exists a number $C(\mathcal{H}) \in \mathbb{N}$ (that depends on $\mathcal{X}$, $\mathcal{Y}$ and $\mathcal{H}$ but does not depend on $n$) such that $M(\mathcal{H}, n) \leq C(\mathcal{H})$.*

The proof of Theorem B.3 uses the following generalization of the Sauer–Shelah–Perles lemma.

**Theorem B.4** ([Nat89]; Corollary 5 in [HL95])**.** *Let $d, n, k \in \mathbb{N}$, let $\mathcal{X}$ and $\mathcal{Y}$ be sets of cardinality $n$ and $k$ respectively, and let $\mathcal{H} \subseteq \mathcal{Y}^{\mathcal{X}}$ such that $\mathsf{ND}(\mathcal{H}) \leq d$. Then*

$$|\mathcal{H}| \leq \sum_{i=0}^{d} \binom{n}{i} \binom{k+1}{2}^i \leq \left(\frac{enk^2}{d}\right)^d.$$

*Proof of Theorem B.3.* Items 1 and 3 and the $\min\{d, n\}$ lower bound in Item 2 follow similarly to the corresponding items in Theorem 4.1. The upper bound in Item 2 also follows similarly to the corresponding item in Theorem 4.1, except that it uses Theorem B.4 instead of the Sauer–Shelah–Perles lemma. The $\lfloor \log(n) \rfloor$ lower bound in Item 2 follows from Theorem D.5 and Claim D.4. $\square$

# C  Combinatorics of Trees

In this section we present a simple lemma from Ramsey theory about trees that is used for proving Theorem D.5. We start with a generalized definition of subtrees.

**Definition C.1.** *Let $X$ be a finite set and let $(X, \preceq)$ be a partial order relation. For $p, c \in X$, we say that $c$ is a child of $p$ if $p \preceq c$ and there does not exist $m \in X$ such that $p \preceq m \preceq c$. We say that $z \in X$ is a leaf if there exists no $x \in X$ such that $z \preceq x$. $(X, \preceq)$ is a binary tree if every non-leaf $x \in X$ has precisely $2$ children. The depth of $z \in X$ is the largest $d \in \mathbb{N}$ for which there exist distinct $x_1, \ldots, x_d \in X$ such that $x_1 \preceq x_2 \preceq \cdots \preceq x_d \preceq z$. For $d \in \mathbb{N}$, we say that $(X, \preceq)$ is a complete binary tree of depth $d$ if $(X, \preceq)$ is a binary tree and all the leaves in $X$ have depth $d$. We say that a partial order $(X', \preceq')$ is a subtree of $(X, \preceq)$ if $X' \subseteq X$, and $\forall a, b \in X' : a \preceq' b \iff a \preceq b$.*

**Lemma C.2** (Lemma 16 in [ALMM19])**.** *Let $p, q \in \mathbb{R}$ be non-negative such that $p + q \in \mathbb{N}$. Let $T = (X, \preceq)$ be a complete binary tree of depth $d = p + q - 1$, and let $f : X \to \{0, 1\}$. Then at least one of the following statements holds:*

- *$T$ has a $0$-monochromatic complete binary subtree of depth at least $p$. Namely, there exists $T' = (X', \preceq')$ such that $T'$ is a subtree of $T$, $T'$ is a complete binary tree of depth at least $p$, and $f(x) = 0$ for all $x \in X'$.*

- *$T$ has a $1$-monochromatic complete binary tree subtree of depth at least $q$.*

For completeness, we include a proof of this lemma.

*Proof of Lemma C.2.* We prove the claim by induction on the depth $d$. The base case of $d = 0$ (a tree with a single node) is immediate. For the induction step, let $a$ denote the root of $T$, and let $T_\ell$

and $T_r$ denote the subtrees of $T$ of depth $d - 1$ consisting of all descendants of the left and right child of $a$ respectively. Assume that $f(a) = 0$. If $T_\ell$ or $T_r$ contain a 1-monochromatic subtree of depth at least $q$, then we are done. Otherwise, by the induction hypothesis, both trees contain a 0-monochromatic subtree of depth at least $p - 1$. Joining these two subtrees as children of the root $a$ yields a 0-monochromatic subtree of depth at least $p$, as desired. The proof for the case $f(a) = 1$ is similar. $\square$

We use the following corollary of Lemma C.2.

**Lemma C.3.** *Let $k, d \in \mathbb{N}$. Let $T = (X, \preceq)$ be a complete binary tree of depth $d \in \mathbb{N}$, and let $f : X \to [k]$. Then $T$ has an $f$-monochromatic complete binary subtree $T' = (X', \preceq')$ of depth at least*

$$d' = \frac{d+1}{2^{\lceil \log(k) \rceil}}.$$

*Namely, there exists $T'$ such that $T'$ is a subtree of $T$, $T'$ is a complete binary tree of depth at least $d'$, and $|\{f(a) : a \in X'\}| = 1$.*

*Proof of Lemma C.3.* We will show that for any $b \in \mathbb{N}$, if $k \leq 2^b$ then there exists an $f$-monochromatic subtree of $T$ of depth at least

$$\frac{d+1}{2^b}.$$

This implies the lemma, which corresponds to the special case $b = \lceil \log(k) \rceil$.

We proceed by induction on $b$. The base case $b = 1$ follows from Lemma C.2. For the induction step, we assume that the claim holds for $b$ and prove that it holds for $b + 1$. Namely, we show that if $f : X \to [k]$ and $k \leq 2^{b+1}$ then there exists an $f$-monochromatic subtree of depth at least $(d+1)/2^{b+1}$.

Define $g : X \to \{1, 2\}$ by $g(x) = 1 + (f(x) \mod 2)$. By Lemma C.2, there exists a $g$-monochromatic complete binary subtree $T_0 = (X_0, \preceq)$ of $T$ of depth at least $(d+1)/2$. In particular $|\{f(x) : x \in X_0\}| \leq 2^b$. By invoking the induction hypotheses on $T_0$, there exists a complete binary subtree of $T_0$ that is $f$-monochromatic and has depth at least

$$\frac{\frac{d+1}{2} + 1}{2^b} > \frac{d+1}{2^{b+1}},$$

as desired. $\square$

## D  Multiclass Threshold Bounds

**Definition D.1.** *Let $\mathcal{X}$ and $\mathcal{Y}$ be sets, let $X = \{x_1, \ldots, x_t\} \subseteq \mathcal{X}$, and let $\mathcal{H} \subseteq \mathcal{Y}^{\mathcal{X}}$. We say that $X$ is threshold-shattered by $\mathcal{H}$ if there exist distinct $y_0, y_1 \in \mathcal{Y}$ and functions $h_1, \ldots, h_t \in \mathcal{H}$ such that $h_i(x_j) = y_{\mathbb{1}(j \leq i)}$. The threshold dimension of $\mathcal{H}$, denoted $\mathsf{TD}(\mathcal{H})$, is the supremum of the set of integers $t$ for which there exists a threshold-shattered set of cardinality $t$.*

We introduce the following generalization of the threshold dimension.

**Definition D.2.** *Let $\mathcal{X}$ and $\mathcal{Y}$ be sets, let $X = \{x_1, \ldots, x_t\} \subseteq \mathcal{X}$, and let $\mathcal{H} \subseteq \mathcal{Y}^{\mathcal{X}}$. We say that $X$ is multi-class threshold-shattered by $\mathcal{H}$ if there exist $y_1, y_1' \ldots, y_t, y_t' \in \mathcal{Y}$ such that $y_i \neq y_j'$ for all $i, j \in [t]$, and there exist functions $h_1, \ldots, h_t \in \mathcal{H}$ such that*

$$h_i(x_j) = \begin{cases} y_i & (j \leq i) \\ y_j' & (j > i). \end{cases}$$

*The multi-class threshold dimension of $\mathcal{H}$, denoted $\mathsf{MTD}(\mathcal{H})$, is the supremum of the set of integers $t$ for which there exists a threshold-shattered set of cardinality $t$.*

See Table 1 for an illustration of this definition.

**Claim D.3.** *Let $\mathcal{X}$ and $\mathcal{Y}$ be sets, $k = |\mathcal{Y}| < \infty$, and let $\mathcal{H} \subseteq \mathcal{Y}^{\mathcal{X}}$. Then $\mathsf{TD}(\mathcal{H}) \geq \lfloor \mathsf{MTD}(\mathcal{H})/k^2 \rfloor$.*

*Proof of Claim D.3.* The proof follows from two applications of the pigeonhole principle. $\square$

| | $x_1$ | $x_1$ | $x_3$ | $x_4$ | $x_5$ |
|---|---|---|---|---|---|
| $h_1$ | $y_1$ | $\mathbf{y}_2'$ | $\mathbf{y}_3'$ | $\mathbf{y}_4'$ | $\mathbf{y}_5'$ |
| $h_2$ | $y_2$ | $y_2$ | $\mathbf{y}_3'$ | $\mathbf{y}_4'$ | $\mathbf{y}_5'$ |
| $h_3$ | $y_3$ | $y_3$ | $y_3$ | $\mathbf{y}_4'$ | $\mathbf{y}_5'$ |
| $h_4$ | $y_4$ | $y_4$ | $y_4$ | $y_4$ | $\mathbf{y}_5'$ |
| $h_5$ | $y_5$ | $y_5$ | $y_5$ | $y_5$ | $y_5$ |

Table 1: An illustration of Definition D.2. The table shows a collection of points $\{x_1, \ldots, x_5\}$ that are multi-class threshold shattered by functions $\{h_1, \ldots, h_5\}$.

**Claim D.4.** *Let $\mathcal{X}$ and $\mathcal{Y}$ be sets, let $\mathcal{H} \subseteq \mathcal{Y}^{\mathcal{X}}$ such that $d = \mathsf{TD}(\mathcal{H}) < \infty$, and let $n \in \mathbb{N}$. Then*

$$M(\mathcal{H}, n) \geq \min \left\{ \lfloor \log(d) \rfloor, \lfloor \log(n) \rfloor \right\}.$$

The proof of Claim D.4 is similar to that of Claim 3.4.

**Theorem D.5.** *Let $\mathcal{X}$ and $\mathcal{Y}$ be sets with $k = |\mathcal{Y}| < \infty$, let $\mathcal{H} \subseteq \mathcal{Y}^{\mathcal{X}}$. If $\mathsf{LD}(\mathcal{H}) = \infty$ then $\mathsf{MTD}(\mathcal{H}) = \infty$.*

*Proof of Theorem D.5.* Let $f_k(d)$ be the largest number such that every class with Littlestone dimension $d$ has multi-class threshold dimension at least $f_k(d)$. We show by induction on $d$ that $f_k$ satisfies the following recurrence relation:

$$f_k(d) \geq \begin{cases} 1 & d = 1 \\ 1 + f_k(\lceil d/2k \rceil - 1) & d > 1 \end{cases}.$$

In particular, this implies that $f_k(d) \xrightarrow{d \to \infty} \infty$, which implies the theorem.

The base case $d = \mathsf{LD}(\mathcal{H}) = 1$ is immediate. For the induction step, we assume the relation holds whenever $\mathsf{LD}(\mathcal{H}) \in [d-1]$, and prove that it holds for $\mathsf{LD}(\mathcal{H}) = d$. Let $T$ be a Littlestone tree of depth $d$ that is shattered by $\mathcal{H}$. Fix $h \in \mathcal{H}$. Then $h$ is a $k$-cloring of the nodes of $T$. By Lemma C.3, there exists an $h$-monochromatic subtree $T' \subseteq T$ of depth at least $\lceil d/2k \rceil$. Let $y$ be the color assigned by $h$ to all nodes of $T'$. $T'$ is shattered by $\mathcal{H}$, so there exists a child $c$ of the root $x$ of $T'$ such that edge from $x$ to $c$ is labeled by some value $y' \neq y$. Let $\mathcal{H}' = \{g \in \mathcal{H} : g(x) = y'\}$. $\mathcal{H}'$ shatters the subtree rooted at $c$, so $\mathsf{LD}(\mathcal{H}') \geq \lceil d/2k \rceil - 1$. By the induction hypothesis, there exist $x_1, \ldots, x_s$ for $s = f_k(\lceil d/2k \rceil - 1)$ that are multi-class threshold shattered by functions $h_1, \ldots, h_s \in \mathcal{H}'$.

By construction, the set $X = \{x_1, \ldots, x_s, x_{s+1} = x\}$ is multi-class threshold shattered by $\{h_1, \ldots, h_s, h_{s+1} = h\}$, because $h_{s+1}(x_j) = y$ for all $j \in [s+1]$, and $h_i(x_{s+1}) = y'$ for all $i \in [s]$. Hence, $f_k(d) \geq s + 1 = 1 + f_k(\lceil d/2k \rceil - 1)$, as desired. $\qquad\square$

# E   Proof of Agnostic Lower Bound

The lower bound in Theorem 6.1 is derived using an anti-concentration technique from Lemma 14 of [BPS09]. Specifically, this technique uses the following inequality.

**Theorem E.1** (Khinchine's inequality; Lemma 8.2 in [CL06]). *Let $k \in \mathbb{N}$, and let $\sigma_1, \sigma_2, \ldots, \sigma_k$ be random variables sampled independently and uniformly at random from $\{\pm 1\}$. Then*

$$\mathbb{E}\left[ \left| \sum_{i \in k} \sigma_k \right| \right] \geq \sqrt{k/2}.$$

*Proof of lower bound in Theorem 6.1.* Let $d = \mathsf{VC}(\mathcal{H})$. Let $\{x_1^*, \ldots, x_d^*\} \subseteq \mathcal{X}$ be a set of cardinality $d$ that is $\mathsf{VC}$-shattered by $\mathcal{H}$. Let $k \in \mathbb{N}$ be the largest integer such that $kd \leq n$.

Let $x \in \mathcal{X}^n$ be a sequence consisting of $k$ copies of the shattered set, namely,

$$(x_1, \ldots, x_{kd}) = \left( x_1^1, x_1^2, \ldots, x_1^k, x_2^1, x_2^2, \ldots, x_2^k, \ldots, x_d^1, x_d^2, \ldots, x_d^k \right),$$

such that $x_i^j = x_i^*$ for all $i \in [d]$ and $j \in [k]$. If $kd < n$ then the remaining $n - kd$ elements of $x$ may be arbitrary.

Consider a randomized adversary that selects the sequence $x$, and chooses all labels to be i.i.d. uniform random bits. For each $i \in [d]$ and $j \in [k]$, let $y_i^j = y_{(i-1)k+j}$ and $\hat{y}_i^j = \hat{y}_{(i-1)k+j}$ denote, respectively, the labels and the predictions corresponding to $x_i^j$. Then for any learner strategy $A$,

$$\mathbb{E}_{y \sim U(\{0,1\}^n)}[R(A, \mathcal{H}, x, y)]$$

$$= \mathbb{E}_{y \sim U(\{0,1\}^n)}\left[\mathbb{E}_{\hat{y}}\left[\sum_{i \in [n]} \mathbb{1}\left(\hat{y}_t \neq y_t\right)\right] - \min_{h \in \mathcal{H}} \sum_{i \in [n]} \mathbb{1}\left(h(x_t) \neq y_t\right)\right]$$

$$= \frac{n}{2} - \mathbb{E}_{y \sim U(\{0,1\}^n)}\left[\min_{h \in \mathcal{H}} \sum_{i \in [n]} \mathbb{1}\left(h(x_t) \neq y_t\right)\right] \qquad (y_i \perp \hat{y}_i)$$

$$\geq \frac{kd}{2} - \mathbb{E}_{y \sim U(\{0,1\}^{kd})}\left[\min_{h \in \mathcal{H}} \sum_{i \in [d]} \sum_{j \in [k]} \mathbb{1}\left(h(x_i^j) \neq y_i^j\right)\right] \qquad (10)$$

$$= \frac{kd}{2} - \mathbb{E}_{y \sim U(\{0,1\}^{kd})}\left[\sum_{i \in [d]} \min_{h \in \mathcal{H}} \sum_{j \in [k]} \mathbb{1}\left(h(x_i^j) \neq y_i^j\right)\right] \qquad (\mathcal{H} \text{ shatters } \{x_1^*, \ldots, x_d^*\})$$

$$= \sum_{i=1}^{d} \frac{k}{2} - \mathbb{E}_{y_i \sim U(\{0,1\}^k)}\left[\min_{h \in \mathcal{H}} \sum_{j \in [k]} \mathbb{1}\left(h(x_i^j) \neq y_i^j\right)\right]$$

$$= \sum_{i=1}^{d} \frac{k}{2} - \mathbb{E}_{y_i \sim U(\{0,1\}^k)}[\min\{r_i, k - r_i\}] \qquad (\text{Let } r_i = \sum_{j \in [k]} y_i^j)$$

$$= \sum_{i=1}^{d} \mathbb{E}_{y_i \sim U(\{0,1\}^k)}\left[\left|\frac{k}{2} - r_i\right|\right]$$

$$= \sum_{i=1}^{d} \mathbb{E}_{y_i \sim U(\{0,1\}^k)}\left[\left|\frac{k}{2} - \sum_{j \in [k]}\left(\frac{1}{2} + \frac{\sigma_i^j}{2}\right)\right|\right] \qquad \left(\text{Let } \sigma_i^j = \begin{cases} 1 & y_i^j = 1 \\ -1 & y_i^j = 0 \end{cases}\right)$$

$$= \frac{1}{2}\sum_{i=1}^{d} \mathbb{E}_{y_i \sim U(\{0,1\}^k)}\left[\left|\sum_{j \in [k]} \sigma_i^j\right|\right]$$

$$\geq \frac{1}{2}\sum_{i=1}^{d} \sqrt{\frac{k}{2}} = \frac{d\sqrt{k}}{2\sqrt{2}} = \Omega\left(\sqrt{nd}\right), \qquad (11)$$

where the final inequality is Khinchine's inequality (Theorem E.1). To see that Inequality (10) holds, let $h^* \in \operatorname{argmin}_{h \in \mathcal{H}} \sum_{i=1}^{kd} \mathbb{1}\left(h(x_t) \neq y_t\right)$, and then

$$\mathbb{E}_{y \sim U(\{0,1\}^n)}\left[\min_{h \in \mathcal{H}} \sum_{i \in [n]} \mathbb{1}\left(h(x_t) \neq y_t\right)\right]$$

$$\leq \mathbb{E}_{y \sim U(\{0,1\}^n)}\left[\sum_{i=1}^{kd} \mathbb{1}\left(h^*(x_t) \neq y_t\right)\right] + \mathbb{E}_{y \sim U(\{0,1\}^n)}\left[\sum_{i=kd+1}^{n} \mathbb{1}\left(h^*(x_t) \neq y_t\right)\right]$$

$$\leq \mathbb{E}_{y \sim U(\{0,1\}^n)}\left[\sum_{i=1}^{kd} \mathbb{1}\left(h^*(x_t) \neq y_t\right)\right] + \frac{n - kd}{2}. \qquad (\{y_t\}_{t > kd} \perp h^*)$$

In particular, Eq. (11) implies that for every learner strategy $A$ there exists $y \in \{0,1\}^n$ such that $R(A, \mathcal{H}, x, y) = \Omega\left(\sqrt{nd}\right)$. $\qquad \square$

