# OpenReview forum: "A Trichotomy for Transductive Online Learning"
_NeurIPS.cc/2023/Conference — NeurIPS 2023 poster_

### Official Review · Reviewer_Hibd · 2023-06-25

**Soundness:** 4 excellent
**Presentation:** 3 good
**Contribution:** 2 fair
**Rating:** 7
**Confidence:** 4

**Summary:**

This paper studies realizable transductive online learning on a fixed and *known* sequence $x_1,\dots,x_n$ (meaning the order is given). The authors state a trichotomy of error rates depending on the finiteness of the VC dimension and Littlestone dimension of the given hypothesis space. Along the way they improve a lower bound on the number of mistakes in terms of the Littlestone dimension. They use the relationship to the threshold dimension to achieve their bounds. Finally, they extend their results to the multi-class case.

**Strengths:**

Nice overview. Well written. Easy to follow proofs.

---- rebuttal comment -----
As authors addressed multiple concerns and added additional results on the agnostic case, I raised my score from 6 to 7. I think this paper is valuable for the online learning community.

**Weaknesses:**

Most of the claimed results are already known or exist implicitly. In particular, all the bounds used in the trichotomy are standard (either from Littlestone's Halving / SOA, or from the online vs offline paper). Even the additional $\log(n)$ lower bound given by the threshold dimension was essentially covered by an example (called "$\sigma_{\text{worst}}$") in [BKM97] (without using the threshold dimension explicitly though). The actual main novelty (which is still a decent and interesting contribution) is the improvement of the lower bound from $\sqrt{\log(LD(H))}$ to $\log(LD(H))$.

The informal version of Thm 4.1. is slightly misleading. E.g., case 2 If $VC(H)=n$ the rate is still $\Theta(n)$ instead of $\log(n)$.

I would suggest writing up the bounds for a finite instance space $|\mathcal{X}|=n$ as corollaries. This might be interesting for various settings (e.g., node classfication) and guide the intuition of the reader.

I am not sure referring to the setting here with the sequence $(x_1,\dots,x_n)$ known in advance to the learner as "transductive" is the best idea. Transductive typically refers to the set (without knowing the order) $\\{x_1,\dots,x_n\\}$ to be known. Even one of the referenced papers [KK05], which the authors use to justify the name change from $M_{worst}$ to transductive, is merely referring the fact that the set is known but not necessarily the sequence (e.g., "the set of unlabeled examples is then presented to the learner") as transductive. Similarly for instance, the paper [CS13] refers to the fact that the set is known (instead of the sequence) as transductive "In this model, we have an arbitrary sequence of
labeled examples $(x_1, y_1), . . . , (x_T , y_T)$, where only the set $\\{x_1, . . . , x_T\\}$ is known". I might be wrong though. Please discuss.

Small typos and writing
* line 126: $u_{\leq k}$ should probably be $x$?
* line 202: "Claim claim"
* line 235: "makes $d$ mistakes --> "makes at most $d$ mistakes".
* line 236: What is $m$? Probably $d$?

Given these weaknesses I would say that this work is rather incremental and has limited novelty. Nevertheless it offers some nice new connections (e.g., threshold dimension) and gives a nice overview (the trichotomy) about bounds which (implicitly or explicitly) exist in previous work.

**Questions:**

Essentially the results (as most such online learning papers) leave the gap between $VC+\log n$ and $VC \cdot \log n$ open. Do you have any ideas how to close this gap? Sure you state that both bounds are tight on specific instance sequences, but maybe there is a different combinatorial parameter (probably depending on the particular sequence?) interpolating between "$+$" and "$\cdot$" allowing to get instance-dependent tight bounds.

You briefly discuss some issues in the multi-class case with $|\mathcal{Y}|=\infty$. However there is the recent work by Brukhim et al [FOCS 2022] using the Daniely-Shalev-Shwartz (DS) dimension to characterize multi-class learnability even if $|\mathcal{Y}|=\infty$. Do these results apply in your setting by swapping the Natarajan dimension (or graph dimension etc.) with the DS dimension?

Is the $\log(LD(H))$ lower bound best possible? I.e., are there specific examples where this is tight (meaning, same upper bound).

**Limitations:**

/

---

> ### Author Rebuttal · Authors · 2023-08-09
>
> **Comment:
> Most of the claimed results are already known or exist implicitly. In particular, all the bounds used in the trichotomy are standard [...]. Even the additional lower bound given by the threshold dimension was essentially covered by an example (called "sigma_worst") in [BKM97] (without using the threshold dimension explicitly though). The actual main novelty (which is still a decent and interesting contribution) is the improvement of the lower bound from
> sqrt(log(LD)) to log(LD).**
>
> **Answer:**
> We believe our paper makes some neat and non-trivial contributions, see our **_General Answer 1_** in the Author Rebuttal section.
>
>
> **Comment:
> The informal version of Thm 4.1. is slightly misleading. E.g., case 2 If $V C(H)=n$ the rate is still $\Theta(n)$ instead of $\log (n)$.**
>
> **Answer:** Following the suggestion of  Reviewer GnLU, we will add a comment that the constants in cases 2 and 3 of the informal trichotomy hide a dependence on the VC and Littlestone dimensions, respectively.
>
> **Comment:
> I would suggest writing up the bounds for a finite instance space $|\mathcal{X}|=n$ as corollaries. This might be interesting for various settings (e.g., node classfication) and guide the intuition of the reader.**
>
> Interesting suggestion! We are happy to do add this to the paper. Are there any references or specific examples that you have in mind? Were you perhaps thinking of something like Herbster, Pontil & Wainer "Online learning over graphs", or something else?
>
> **Comment:
> I am not sure referring to the setting here with the sequence $\left(x_1, \ldots, x_n\right)$ known in advance to the learner as "transductive" is the best idea. Transductive typically refers to the set (without knowing the order) $\\{x_1, \dots, x_n\\}$ to be known.**
>
> **Answer:** Thank you for pointing this out! There appears to be some ambiguity in the literature regarding which of these settings should be called "transductive". For instance, Syrgkanis, Krishnamurthy and Schapire (http://proceedings.mlr.press/v48/syrgkanis16.pdf) write of "a transductive setting (Ben-David et al., 1997) in which the learner knows the arriving contexts a priori, or, less stringently, knows only the set, but not necessarily the actual sequence or multiplicity with which each context arrives." We agree that it is better to avoid this ambiguity. To that end, we will qualify the name used in our paper, distinguishing between a "sequence-transductive setting" and "set-transductive setting" or something similar. Suggestions for names are welcome!
>
> **Comment:
> Essentially the results (as most such online learning papers) leave the gap between $VC+\log n$ and $VC \cdot \log n$ open. Do you have any ideas how to close this gap? Sure you state that both bounds are tight on specific instance sequences, but maybe there is a different combinatorial parameter (probably depending on the particular sequence?) interpolating between "+" and "\cdot" allowing to get instance-dependent tight bounds.**
>
> We agree that this is an intriguing question. We don't know the answer, and we intend to continue investigating this in future worsks.
>
> **Comment:
> You briefly discuss some issues in the multi-class case with $|\mathcal{Y}|=\infty$. However there is the recent work by Brukhim et al using the DS dimension to characterize multi-class learnability even if $|\mathcal{Y}|=\infty$. Do these results apply in your setting by swapping the Natarajan dimension with the DS dimension?**
>
> **Answer:**
>
> Excellent question! It is indeed interesting to note that the **_DS dimension does not characterize learnability_** in the transductive online learning setting. Here is a counterexample: for any natural $n$, we construct a hypothesis class $\mathcal{H}$ that has DS dimension $1$, but the adversary can force $M(\mathcal{H},n)=n$ mistakes. The class is defined as follows. Let $\{x_0,\dots,x_{n-1}\}\subseteq \mathcal{X}$ be distinct instances. Consider a complete binary tree of depth $n$, such that for each $i \in \{0,1,...,n-1\}$, all the nodes at layer $i$ of the tree (i.e., at distance $i$ from the root) are labeled with instance $x_i$, and each edge in the tree is labeled with a unique label (that does not appear anywhere else in the tree). Let $\mathcal{H}$ be a set functions that shatters this tree. Observe that the adversary can select the sequence $(x_0,\dots,x_{n-1})$ and force $n$ mistakes. We now argue that the DS dimension of $\mathcal{H}$ is $1$.
>
> Assume for contradiction that there exists a $2$-pseudocube for $\mathcal{H}$ in the DS sense. Namely, there exists a collection of vectors of length 2, $C = \{((x_i^1,y_i^1),(x_i^2,y_i^2)): i \in I\} \subseteq (\mathcal{X}\times\mathcal{Y})^2$ such that each vector is realizable by a function in $\mathcal{H}$, and for each $v = ((x_i^1,y_i^1),(x_i^2,y_i^2)) \in C$ and $j \in \{0,1\}$ there exists $\tilde{v} = ((x_i^1,\tilde{y}_i^1),(x_i^2,\tilde{y}_i^2)) \in C$ such that $y^j_i = {\tilde{y}_i}^j$ and $y_i^{1-j} \neq \tilde{y}_i^{1-j}$.
>
> Fix $j$ and a pair $v, \tilde{v}$ that agree on the label of $x_i^j$ and disagree on the label of $x_i^{1-j}$. Choose $v$ and $\tilde{v}$ that $x_i^j$ appears deeper down in the tree than $x_i^{1-j}$ does (this is possible because $C$ is a pseudocube). Because all the labels in the tree are unique, if two functions agree on the label for the deeper node $x_i^j$, then they must also agree on the label for the less deep node $x_i^{1-j}$. This is a contradiction to the assumption that both $v$ and $\tilde{v}$ are realizable by $\mathcal{H}$.
>
> **Comment:
> Is the log(LD) lower bound best possible? I.e., are there specific examples where this is tight (meaning, same upper bound).**
>
> Excellent question. We have an idea that could potentially yield an $\sqrt{\mathsf{LD}}$ upper bound, but it is quite complex and would justify an entirely new paper (if it works). Even if that proof works, the gap between the upper and lower bounds would still be large.

---

> > ### Comment · Reviewer_Hibd · 2023-08-14
> >
> > Thank you for your clarifications.
> >
> > Interesting that the DS dimension does not characterise this setting here!
> >
> > I like the terms set-transductive vs sequence-transductive. Either way, some paragraph on related work discussing this ambiguity would be more than sufficient and helpful for readers.
> >
> > Do you think that your derivation of the $log(TD(H))$ lower bound is essentially using the same arguments as the one discussed for the specific sequence $\sigma_\text{worst}$ in BKM97? Please comment.
> >
> > Another small comment. Maybe indicate in Thm 4.1 that $C(H)\leq d$ (with $d=Ldim$) otherwise it is unclear how large this quantity might be without looking at the proof.
> >
> > Finally, I'd be happy if you can give more comments on the agnostic setting mentioned by another reviewer. Do you have some first results?

---

> > > ### Author Response · Authors · 2023-08-16
> > >
> > > Thank you for your interest! We have added a detailed discussion on the agnostic case, please see **_General Answer 2: The Agnostic Case_** above.
> > >
> > > We will address your remaining points soon :)

---

> > > > ### Comment · Reviewer_Hibd · 2023-08-16
> > > >
> > > > Thank you I raised my score to a 7.
> > > >
> > > > And yes with online node classification / online learning on graphs I've mean papers like
> > > >
> > > > * N. Cesa-Bianchi, C. Gentile, F. Vitale, G. Zappella
> > > > Random Spanning Trees and the Prediction of Weighted Graphs (JMLR 2013)
> > > > * Mark Herbster, Stephen Pasteris, Shaona Ghosh "Online prediction at the limit of zero temperature" (NeurIPS 2015)
> > > >
> > > > and the one you mentioned (and other related papers)

---

> > > > > ### Author Response · Authors · 2023-08-18
> > > > >
> > > > > That is wonderful! We are thrilled to hear this! :)

---

> > > > > > ### Author Response · Authors · 2023-08-18
> > > > > >
> > > > > > Briefly addressing remaining comments:
> > > > > >
> > > > > > **1. Do you think that your derivation of the lower bound is essentially using the same arguments as the one discussed for the specific sequence $\sigma_{\text{worst}}$ in BKM97? Please comment.**
> > > > > >
> > > > > > Yes, Claim 3.4 in our paper generalizes the example from BKM97.
> > > > > >
> > > > > > Edit: To clarify, that part of our argument is similar to that example. However, our proof of the $\log(n)$ lower bound in the trichotomy also uses the threshold dimension and the connection between threshold dimension and Ldim, which are novel in this context.
> > > > > >
> > > > > > **2. Maybe indicate in Thm 4.1 that $C(H) < d$ (with $d = Ldim$) otherwise it is unclear how large this quantity might be without looking at the proof.**
> > > > > >
> > > > > > Yes, this is a good suggestion. We will add this in the final version.
> > > > > >
> > > > > > ___
> > > > > >
> > > > > > Thank you for your time and careful reading of our paper!

---

### Official Review · Reviewer_GnLU · 2023-07-06

**Soundness:** 4 excellent
**Presentation:** 4 excellent
**Contribution:** 2 fair
**Rating:** 5
**Confidence:** 3

**Summary:**

This paper studies transductive online learning, which differs from standard learning in that the adversary must commit to a sequence of instances to be labelled by the learner at the start of the game. The adversary's strategy can thus only be adaptive w.r.t. the labeling of the sequence, not the sequence itself. The goal of the paper is to bound the number of mistakes made by learning algorithms on a sequence of length $n$.

The main result is comprised of three different set ups:
- If the VC dimension of a concept class is infinite, then the number of mistakes is $n$,
- If the VC dimension is finite, but the Littlestone dimension is infinite, then the number of mistakes is logarithmic in $n$,
- Finally, if the Littlestone dimension is finite, then the number of mistakes is constant (in the sense that it does not depend on $n$ -- it will however depend on the Littlestone dimension).

**Strengths:**

- Conceptually interesting to have a trichotomy that depends on the VC and Littlestone dimensions
- The paper is well-written and easy to follow, especially the proofs, which are clear and well-explained
- The proof of Theorem 3.1 is interesting, and carries the technical weight of the paper, in addition to being a significant improvement on previous lower bounds
- This paper seems of interest to the online learning theory community

**Weaknesses:**

- Apart from the proof of Theorem 3.1, the proofs seem relatively straightforward, so the originality/novelty of the work is mainly conceptual, not technical. More specifically, items 1 and 3 from Theorem 4.1 are applications of the definitions of the VC and Littlestone dimensions -- but Theorem 4.1 is the main contribution of the paper, apart from the lower bound. Item 2 on appearance seems more substantial but l.218-225 are simply explaining the Halving algorithm, leaving the main technical tool to be an application of the Sauer-Shelah-Perles lemma.
- A substantial improvement would come from getting a tighter bound w.r.t. the Littlestone dimension for the constant in item 3 of Theorem 4.1, as for now the proof (l.234-236) follows directly from the definition of the Littlestone dimension (of course, easier said than done! I just think there is some technical weight missing in the paper)

I was hesitating between a 4 and 5, but opted for 5 because of Thm 3.1 and the conceptual contribution.

**Questions:**

Questions
- l.236: what is $m$? Do you mean $d$?
- Is there a fundamental difference in the derivation of the results for the multiclass setting (vs binary)?

Comments:
- Informal version of Thm 4.1 (l.69): I think it should be specified that the constants in cases 2 and 3 hide a dependence on the VC and Littlestone dimensions, respectively.
- l.202: "Claim claim"
- Perhaps include a future work/conclusion section?

---

> ### Author Rebuttal · Authors · 2023-08-09
>
> **Comment:**
> **Apart from the proof of Theorem 3.1, the proofs seem relatively straightforward, so the originality/novelty of the work is mainly conceptual, not technical. [...] I just think there is some technical weight missing in the paper.**
>
> **Answer:**
> We believe our paper makes some neat and non-trivial contributions, see our **_General Answer 1_** in the Author Rebuttal section.
>
>
> **Comment: l.236: what is m? Do you mean d?**
>
> **Answer:** Yes, this should be $d$. Thanks for identifying this typo!
>
> **Question:**
> **Is there a fundamental difference in the derivation of the results for the multiclass setting (vs binary)?**
>
> **Answer:**
> Technically, the multi-class case involves some considerable additional work, as we explain in the third bullet in _General Answer 1_ in the Author Rebuttal section.
>
> **Comment:**
> **Informal version of Thm 4.1 (l.69): I think it should be specified that the constants in cases 2 and 3 hide a dependence on the VC and Littlestone dimensions, respectively.**
>
> **Answer:**
> Thanks for the suggestion! We will add this.
>
> **Question:**
> **Perhaps include a future work/conclusion section?**
>
> **Answer:**
> The camera-ready version will include a Future Works section, touching on the agnostic case, and discussing our thoughts on obtaining sharper bounds in the $\Theta(\log(n))$ and $\Theta(1)$ cases of the trichotomy.

---

> > ### Comment · Reviewer_GnLU · 2023-08-11
> > **Response**
> >
> > Thanks for the general reply and for the specific answers to my questions. I think it would be worth highlighting the technical contribution of the multi-class setting in the main paper, even at a high level -- otherwise it seems like there isn't much difference with the binary setting. Given the page limit, there is definitely enough room to add this discussion/overview!

---

> > > ### Comment · Reviewer_Hibd · 2023-08-14
> > >
> > > Dear reviewer GnLU, dear authors,
> > > I'd really like to see more comments on the agnostic case. Do the authors have first results?
> > > In standard online learning (without the sequence known in advance to the learner) tight results exist (see e.g., [1,2] and perhaps some more). Do these apply here?
> > >
> > > Best,
> > > Hibd
> > >
> > > [1] Ben-David, Shai, Dávid Pál, and Shai Shalev-Shwartz. "Agnostic Online Learning." COLT 2009
> > > [2] Hanneke, Steve, et al. "Multiclass online learning and uniform convergence. COLT 2023

---

> > > > ### Author Response · Authors · 2023-08-16
> > > >
> > > > Thank you for your interest! We have added a detailed discussion, please see **_General Answer 2: The Agnostic Case_** above.

---

> > > > > ### Author Response · Authors · 2023-08-18
> > > > >
> > > > > Also, following your comment, we will be sure to highlight the technical contribution of the multi-class setting in the final version of the paper :)

---

### Official Review · Reviewer_e1Y3 · 2023-07-09

**Soundness:** 3 good
**Presentation:** 2 fair
**Contribution:** 1 poor
**Rating:** 6
**Confidence:** 3

**Summary:**

The paper considers the realizable case in the transductive online learning setting. It shows that, for a sequence of length $n$, the number of errors is $\Theta(n)$, $\Theta(\log n)$ or $\Theta(1)$ depending on the finiteness of the VC dimension and the Littlestone dimension. In the last case, the paper also improves the dependence on the Littlestone dimension $D_L$ from $\Omega(\sqrt{D_L})$ to $\Omega(D_L)$.

**Strengths:**

The paper looks sound and their main result is clear.

**Weaknesses:**

I have the following concerns:
1. The significance of the main contribution. In Theorem 4.1, the only non-trivial case is the upper bound in Item 2, and, as I understand, it follows from [KK05]. Maybe pointing out the trichotomy is significant in itself, but I'm not in the area, so I can't judge it.
2. Presentation should be improved. It took me some time to parse Definition 2.4, and I could do this only because I knew the definition before.  Both this definition and the reasoning in lines 185-188 are better replaced with pictures.
3. I didn't judge by this point, but I suspect that not all relevant references are included. E.g. I think that some papers [KK05] should potentially be cited.

Minor issues:
* Line 202: "claim" is repeated
* Why did you move some material to the supplementary? There should be enough space in the main body.

**Questions:**

* Could the multi-dimensional case be stated in terms of graph dimension? Then there would be no $\log k$ factor.

**Limitations:**

The paper only considers the realizable case

---

> ### Author Rebuttal · Authors · 2023-08-09
>
> **Comment:**
>
> **The significance of the main contribution. In Theorem 4.1, the only non-trivial case is the upper bound in Item 2, and, as I understand, it follows from [KK05]. Maybe pointing out the trichotomy is significant in itself, but I'm not in the area, so I can't judge it.**
>
> **Answer:**
>
> Please see General Answer 1 above. Regarding [KK05], note that:
>  - They do not discuss lower bounds
>  - Their upper bounds are a bit different. Specifically, they use a randomized or “hallucination” algorithm to show an expected bound whereas we show a worst-case bound (we also shave off a log(d) factor from their result, reducing from $d\log(n))$ to $d\log(n/d))$, but that is easy).
>
>
> **Comment:**
> **Presentation should be improved. It took me some time to parse Definition 2.4, and I could do this only because I knew the definition before. Both this definition and the reasoning in lines 185-188 are better replaced with pictures.**
>
> **Answer:**
>
> Agreed, we will add figures for the definition of a Littlestone tree and for the dyadic order argument.
>
> **Comment:**
> **I think that some papers [KK05] should potentially be cited.**
>
> **Answer:**
>
> Thank you for pointing this out! The camera-ready version will include a discussion comparing our work with [KK05] and the other papers mentioned in Footnote 1 on Page 1.

---

> > ### Comment · Reviewer_e1Y3 · 2023-08-13
> >
> > Thank you for the reply. Can you please answer the question about the graph dimension?

---

> > > ### Author Response · Authors · 2023-08-14
> > >
> > > Thank you for calling this to our attention! Apologies, we accidentally omitted our answer to the question about the graph dimension.
> > >
> > > We actually don't see a reason why using the graph dimension would allow us to eliminate the $\log(k)$ factor from the upper bound in Theorem B.2. Our upper bound uses the Sauer–Shelah–Perles lemma for the Natarajan dimension, which states that a class with Natarajan dimension $d$ and $k$ possible labels has at most $\left(e n k^2/d\right)^d$ functions, so the halving algorithm will make at most $O(d \log (n k / d))$ mistakes.
> > >
> > > Similarly, note that a class with graph dimension $d$ can also have a number of functions that grows roughly like $k^d$, so the same upper bound would follow if we apply the same proof strategy to the graph dimension. For example, the class of all functions $f: [d] \to [k]$ has graph dimension $d$ and contains $k^d$ functions.
> > >
> > > While this shows that the same proof would not yield an improved upper bound, it is of course entirely possible that a different proof might do so.

---

> > > > ### Author Response · Authors · 2023-08-18
> > > >
> > > > Dear Reviewer e1Y3,
> > > >
> > > > In your review, you mentioned the following points:
> > > >
> > > > * **"The significance of the main contribution. In Theorem 4.1, the only non-trivial case is the upper bound in Item 2, and, as I understand, it follows from [KK05]."** We feel this is not accurate. Please see **[General Answer 1](https://openreview.net/forum?id=iSd8g75QvP&noteId=wuoq32YtUM)**.
> > > >
> > > > * **"Could the multi-dimensional case be stated in terms of graph dimension?"** We have answered that question.
> > > >
> > > > * **You commented on related works.** In the final version, we will expand our discussion of related works, including the paper [KK05] that you mentioned.
> > > >
> > > > * **"The paper only considers the realizable case."** Following your comment, we have added a fairly complete analysis of the agnostic case. Please see **[General Answer 2: The Agnostic Case](https://openreview.net/forum?id=iSd8g75QvP&noteId=NpnLZNlj4s)**.
> > > >
> > > > Thus, we feel that we have fully addressed all the concerns raised in your review. Additionally, there has been some significant discussion with the other reviewers. In particular, we invite you to read **[this very nice summary by Reviewer A8Lo](https://openreview.net/forum?id=iSd8g75QvP&noteId=oTP7gwPBR2)**.
> > > >
> > > > At this point, do you have any major concerns about our paper? If not, would you be willing to reconsider your score for our paper?

---

> > > > > ### Comment · Reviewer_e1Y3 · 2023-08-18
> > > > >
> > > > > Thank you for the replies. I agree that the agnostic case makes the paper stronger. After all the clarifications, I do believe that the paper has enough technical weight, and I raised my score.

---

> > > > > > ### Author Response · Authors · 2023-08-20
> > > > > >
> > > > > > That's wonderful! We are thrilled to hear this! :)

---

### Official Review · Reviewer_gZ3U · 2023-07-14

**Soundness:** 4 excellent
**Presentation:** 4 excellent
**Contribution:** 3 good
**Rating:** 5
**Confidence:** 4

**Summary:**

This work considers transductive online learning where the pool of examples to be labeled is fixed and known to the learner.  Thus, the adversary can
control the order that the points are presented to the learner and the label that the learner receives in each round.  The  goal of the learner is to predict labels in each round so that they do as few mistakes as possible.  In this work the
authors provide new results on the minimum possible number of mistakes.
They show that the minimal mistakes can either be constant, grow logarithmically, or grow linearly with the number of examples $n$ that are to be labeled by the learner.  More precisely, they show that if the VC dimension of the class is infinite then the number of mistakes is $n$, if the VC dimension is bounded but the Littlestone dimension is infinite then the number of mistakes is $\Theta(\log n)$, and when the Littlestone dimension is bounded the number of mistakes is constant.  Morever, when the Littlestone dimension is at most $d$ the authors show a $\Omega(\log d)$ lower bound on the number of mistakes.


**Strengths:**

The transductive learning model is a well-motivated, and interesting learning model that may often be more realistic than the fully adversarial (standard) online setting where the adversary can choose the examples to be labeled.

This work provides a new trichotomy result and an improved bound ($\Omega(\log d)$ instead of $\Omega(\sqrt{\log d})$ that was shown in the prior work) on the number of mistakes under bounded Littlestone dimension. Moreover, the argument for the $\Omega(\log d)$ is nice.

**Weaknesses:**


My main concern about the main result (the trichotomy of Theorem 4.1) of this work follows rather easily from prior works.  It is not hard to show that when the VC dimension is infinite the adversary can always make the learner to do $n$ mistakes.  Moreover, with bounded VC dimension (less than or equal to $d$), the algorithm that achieves the $d \log (n/d)$ mistake bound is the standard halving algorithm that has been used extensively in the online learning literature.  Finally for the third assertion of Theorem 4.1, when Littlestone dimension is bounded the work of Littlestone '87 directly yields a finite mistake bound. The multiclass generalization of the trichotomy is a nice generalization but follows in the same way as the binary result and does not seem to add much to the paper from a technical point of view.

**Questions:**

See weaknesses. Stating the key ideas or differences of the techniques of this work compared to the prior works could help.

**Limitations:**

Yes.

---

> ### Author Rebuttal · Authors · 2023-08-09
>
> **Comment:
> My main concern about the main result (the trichotomy of Theorem 4.1) of this work follows rather easily from prior works. It is not hard to show that when the VC dimension is infinite the adversary can always make the learner to do
>  mistakes. Moreover, with bounded VC dimension (less than or equal to $d$), the algorithm that achieves the $d\log(n/d)$ mistake bound is the standard halving algorithm that has been used extensively in the online learning literature. Finally for the third assertion of Theorem 4.1, when Littlestone dimension is bounded the work of Littlestone '87 directly yields a finite mistake bound.**
>
> **Answer:**
>
> We believe our paper makes some neat and non-trivial contributions, and is worthy of publication. We invite the reviewer to reconsider their position on this after reading our **_General Answer 1_** in the Author Rebuttal section.
>
>
> **Comment:
> The multiclass generalization of the trichotomy is a nice generalization but follows in the same way as the binary result and does not seem to add much to the paper from a technical point of view.**
>
> **Answer:**
>
> Specifically, see the third bullet of General Answer 1 in the Author Rebuttal.
>
> **Comment:**
>
> **Stating the key ideas or differences of the techniques of this work compared to the prior works could help.**
>
> **Answer:**
>
> The camera-ready version will include a discussion comparing our work with [KK05] and the other papers mentioned in Footnote 1 on Page 1.

---

> > ### Author Response · Authors · 2023-08-16
> >
> > Please see our **_General Answer 2: The Agnostic Case_** above. We feel this is a nice addition!

---

> > > ### Comment · Reviewer_gZ3U · 2023-08-17
> > > **Thanks for the Response**
> > >
> > > I would like to thank the authors for their response. I agree that the agnostic case is a nice addition which makes the story better.  I am somewhat skeptical about adding new results post submission but I am willing to vote for acceptance (I will raise my score to 5).

---

> > > > ### Author Response · Authors · 2023-08-18
> > > >
> > > > Thanks! This is very much appreciated! :)

---

### Official Review · Reviewer_a8Lo · 2023-07-23

**Soundness:** 3 good
**Presentation:** 3 good
**Contribution:** 3 good
**Rating:** 7
**Confidence:** 4

**Summary:**

This paper studies transductive online learning. In this setup, the adversary fixes the sequence of unlabeled instances and reveals labels sequentially following learner's prediction in each round. The paper shows a trichotomy of rates for binary classification: $n$, $\Theta(\log{n})$ and $\Theta(1)$ based on finiteness/infiniteness of VC and Littlestone dimension of the hypotheses class. The authors also extend the result to multiclass setting establishing a similar trichotomy depending on Natarajan and multiclass extension of Littlestone dimension. Finally, the authors provide a quadratic improvement in the known lower bound in the case $\Theta(1)$.

**Strengths:**

1. The paper identifies the trichotomy of rates in online transductive learning. Such trichotomy has been identified in the recently introduced setting of universal learning [BHMvHY2021]. As far as I know, this is the first paper to note such trichotomy in any variant of online learning model. Qualitatively, the paper shows that VC, instead of more restrictive Littlestone, is necessary and sufficient for  learnability of the hypotheses class in transductive online setting.

2. The paper studies a theoretically interesting problem and motivates the problem through a meaningful example. Apart from the proof in Section 6 which I found to be written sloppily, the paper is well-written and easy to follow.

3. The proof technique of Theorem 3.1 is novel and differs from standard lower bound techniques in online learning literature.



[BHMvHY2021]  Olivier Bousquet, Steve Hanneke, Shay Moran, Ramon van Handel, and Amir Yehudayoff. 2021. A theory of universal learning. In Proceedings of the 53rd Annual ACM SIGACT Symposium on Theory of Computing (STOC 2021). Association for Computing Machinery, New York, NY, USA, 532–541. https://doi.org/10.1145/3406325.3451087


**Weaknesses:**

My general impression of the paper is that there is a room for a more comprehensive treatment of the material considered here.

1. I might be speaking with a hindsight bias here, but I think most of proofs regarding trichotomy (Theorem 4.1) is straightforward except that of Claim 3.4 (also see my question below).  In view of that, I would have preferred to see a sharper analysis of mistake bounds in Items 2 and 3 . Namely: (a)  What is the optimal dependence of  $\text{VC}(\mathcal{H})$ and $n$ for the case $\text{VC}(\mathcal{H}) < \infty$ and $\text{LD} (\mathcal{H}) = \infty$? (b) Is there an algorithm that exploits the transductive nature of the game to get a mistake bound better than $\text{LD}(\mathcal{H})$ when it is finite?

2. The exposition of proof of Theorem 3.1, arguably the main technical contributions of the paper, can be improved. For instance, what is $j$ in $\varepsilon_t = \varepsilon^{(j+1)}$ in line 267? What is the role of the constant $C_1$ in the proof/result? Line 263 claims we can take $C_1=5$? If so, why not just use  $5$ for a clearer presentation (although I dont see why it cant be 2)? What is superscript $1$ in $C_1$ indexing? The paragraph following line 270 (that analyzes the adversary) can also be improved in its clarity.

3. The authors could have considered agnostic setting as well. What is the characterization of learnability for online agnostic transductive learning? Does one observe such trichotomy of rates?





**Questions:**

1. Does Theorem 3.1 imply the $\log{n}$ lower bound in Item (2) of Theorem 4.1 using the fact that $\text{LD}(\mathcal{H}) \geq n$ for each $n$? If so, is the discussion of Threshold dimension and Claim 3.4 required to establish the trichotomy for binary case? What about multiclass extension?

2. In Line 44-45, the authors claim that neither party benefit  from using randomness. Do authors mean that there wont be any quantitative difference (up to a constant factor) between deterministic and randomized learning rules? This is most likely true but I would like authors to justify their statement instead of just claiming it.


**Limitations:**

Not applicable.

---

> ### Author Rebuttal · Authors · 2023-08-09
>
> **Comment:
> I might be speaking with a hindsight bias here, but I think most of proofs regarding trichotomy (Theorem 4.1) is straightforward except that of Claim 3.4 (also see my question below).**
>
> **Answer:** We believe our paper makes some neat and non-trivial contributions, see our **_General Answer 1_** in the Author Rebuttal section.
>
> **Comment**
> **The exposition of proof of Theorem 3.1, arguably the main technical contributions of the paper, can be improved.**
>
> **Answer:** Agreed. That proof is clearly less polished than the rest of the paper. In a camera-ready version, we would be sure to iron it out and bring it up to the standard of writing we have shown in the rest of the paper. (The math of the proof is correc)
>
> **Comment:
> Does Theorem 3.1 imply the $\log n$ lower bound in Item (2) of Theorem 4.1 using the fact that $\mathsf{LD}(\mathcal{H}) \geq n$ for each $n$ ? If so, is the discussion of Threshold dimension and Claim 3.4 required to establish the trichotomy for binary case? What about multiclass extension?**
>
> **Answer:** This is a good question! But no -- Theorem 3.1 does not imply the $\log(n)$ lower bound in the trichotomy. Suppose that $\mathsf{LD}(\mathcal{H}) \geq d$ for some integer $d$. In the lower bound of Theorem 3.1, the adversary takes a shattered Littlestone tree of depth $d$, presents the nodes of that tree in breadth-first order, and forces $c_0 \cdot \log(d)$ mistakes. However, a tree of depth $d$ has $\Omega(2^d)$ nodes, so the lower bound implies that if the adversary presents $n = \Omega(2^d)$ instances, it can force $c_0 \cdot \log(d) = \Omega(\log\log(n))$ mistakes. Namely, using Theorem 3.1 would yield a lower bound of $\Omega(\log\log(n))$ mistakes, which is exponentially weaker than the $\log(n)$ lower bound we present in Item (2) of Theorem 4.1.
>
> As an aside, we note that beyond this quantitative advantage, our proof of the $\log(n)$ lower bound via the threshold dimension has the additional advantage that it generalizes to the multiclass setting, as we show in the supplementary materials.
>
> **Question:**
>
> **Is there an algorithm that exploits the transductive nature of the game to get a mistake bound better than
>  when it is finite?**
>
> **Answer:**
> This is an open question. The "correct" mistake bound in the $\Theta(1)$ case can be anywhere between $\mathsf{LD}(\mathcal{H})$ and $\log(\mathsf{LD}(\mathcal{H}))$.
>
> **Question:**
>
> **In Line 44-45, the authors claim that neither party benefit from using randomness. Do authors mean that there wont be any quantitative difference (up to a constant factor) between deterministic and randomized learning rules? This is most likely true but I would like authors to justify their statement instead of just claiming it.**
>
> **Answer:**
> Agreed. This is not hard, we will include a proof in the camera-ready version.

---

> > ### Comment · Reviewer_a8Lo · 2023-08-11
> >
> > Thank you for your general reply and for answering my specific question. I look forward to reading the polished version of the proof, and I think it would be helpful to include more discussion on proof techniques for multiclass settings in the main text. Given the response by the authors, I am happy to raise the score to 5.

---

> > > ### Author Response · Authors · 2023-08-16
> > >
> > > We are thrilled to hear that you are raising your score!!
> > >
> > > Per your question on the agnostic case:
> > >
> > > **The authors could have considered agnostic setting as well. What is the characterization of learnability for online agnostic transductive learning? Does one observe such trichotomy of rates?**
> > >
> > > Please see our **_General Answer 2: The Agnostic Case_** above. We feel this is a valuable addition!

---

> > > > ### Comment · Reviewer_a8Lo · 2023-08-16
> > > > **Reply to authors**
> > > >
> > > > Thank you for providing details about the agnostic case. Although the lower-bound proof uses a standard technique form [1] and the upperbound proof involves a standard Sauer-Selah-Pereles argument, I think the result makes the overall story complete. Also, I do not think proof techniques being standard is a weakness of the paper, but rather the strength of our tools. Here is my revised assessment of the paper after the response by the authors:
> > > >
> > > > 1. Identifying the trichotomy was already an important contribution of this paper.
> > > >
> > > > 1. The qualitative characterization of learnability in this setting is interesting and overall complete. Just to reiterate, a binary hypothesis class $\mathcal{H}$ is learnable in a transductive setting if and only if $ \text{VC}(\mathcal{H}) < \infty$. And the rates involving $\text{VC}(\mathcal{H})$ are very reasonable. As we all know, removing that $\sqrt{\log{T}}$ is typically difficult in the online setting.
> > > >
> > > > 2. Correct me if I am wrong, but your argument for the agnostic case should easily extend to a multiclass setting with finite $|\mathcal{Y}|$ as well, right? One can consider a Natarajan shattered set and sample uniformly between labels of two witness functions for each x for the lower bound, and use multiclass Sauer-Selah-Pereles for the upper bound.
> > > >
> > > > 3. Next, an interesting question is what happens in the case where $|\mathcal{Y}|$ is unbounded? Your example in the paper showing that the analogy breaks down is interesting. However, what I find more interesting is your answer to reviewer Hibd, which shows that the DS dimension does not characterize learnability in the transductive setting when  $|\mathcal{Y}| $ is unbounded. Finding a dimension that characterizes learnability in the transductive setting when $|\mathcal{Y}| $ is unbounded is an exciting open problem.
> > > >
> > > > I raised my score from 4 to 5 because the authors addressed my concerns regarding the necessity of threshold dimension and the proof of the lower bound of $\log{n}$. Given that they also worked out the details for the agnostic case (one of the points of weakness in my original assessment)and provided a very interesting example (I would like authors to include this example in the paper) showing DS dimension does not characterize learnability makes this paper complete and strong in my opinion. Following these modifications and addition, I think this will be an important contribution to the field and should be accepted.
> > > >
> > > > In view of my revised assessment, I am raising my score to 7 and my confidence back to 4.  Also, I am willing to strongly vouch for the acceptance of this paper.
> > > >
> > > >
> > > >
> > > >
> > > > [1] Shai Ben-David, Dávid Pál, and Shai Shalev-Shwartz. Agnostic Online Learning. COLT 2009.

---

> > > > > ### Author Response · Authors · 2023-08-18
> > > > >
> > > > > That is wonderful! We are thrilled to hear this! :)
> > > > >
> > > > > We will be sure to include the example regarding the DS dimension in the final version. I believe you are correct about extending the agnostic arguments to the case of finite $k = |\mathcal Y|$ and we will include that in the final version as well!

---

### Author Rebuttal · Authors · 2023-08-09

**General Comment 1:**

**A recurring comment is that the technical contribution of the paper is not substantial enough (e.g., Reviewer gZ3U: “the main result of this work follows rather easily from prior works”).**

**General Answer 1:**
 -  First, we believe that our trichotomy (Theorem 4.1) is a meaningful conceptual contribution. As one reviewer pointed out, “Such trichotomy has been identified in the recently introduced setting of universal learning [BHMvHY2021] [...] this is the first paper to note such trichotomy in any variant of online learning model”. We feel that this result uncovers an elegant picture of the landscape. We believe it is neat and didactic enough that it could, for instance, be taught in an undergraduate course that covers online learning. The community would benefit from having this published.
 - That being said, the proof of the trichotomy does include a technical contribution that is not entirely trivial. Specifically, the lower bound in item 2 of the trichotomy uses the lower bound that follows from the threshold dimension (Theorem 3.3 and Claim 3.4). One reviewer called this a “nice new connection”.
 - On top of that, the proof of the multiclass trichotomy (Theorem B.2 in the supplementary materials) involves considerably more technical work. First, it requires identifying the “correct” definition of threshold dimension for the multiclass case (Definition A.2). Next, the proof of the log(n) lower bound (Theorem A.5) uses a result from Ramsey theory (Lemma A.7), which would probably not be the first thing that comes to mind for most people. We did not emphasize this technical aspect in the main body of the paper because we focused on the conceptually-clean message of the trichotomy, but we feel that the appendix constitutes a worthy technical contribution.
 - Finally, the proof of our $\log(\mathsf{LD}(\mathcal{H}))$ lower bound (Theorem 3.1) is both non-trivial, and improves on the previous result of Ben-David, Kushilevitz, and Mansour from 26 years ago. As one reviewer wrote, “the proof technique of Theorem 3.1 is novel and differs from standard lower bound techniques in online learning literature.” We maintain that this result does not follow “rather easily” from prior works.

Taken together, we believe that these contributions merit accepting the paper to NeurIPS.

---

> ### Author Response · Authors · 2023-08-16
> **The Agnostic Case**
>
>
> **General Answer 2: The Agnostic Case**
>
> A number of reviewers have asked for our thoughts concerning the agnostic case. Let us briefly review what is known, and summarize our initial observations:
>
> In the _standard_ (non-transductive) agnostic online setting, Ben-David et al. [1] showed that $R_T(\mathcal{H})$, the agnostic online regret for a hypothesis class $\mathcal{H}$ with Littlestone dimension $d_L$ for horizon size $T$ satisfies
>
> $\Omega(\sqrt{d_L \cdot T}) \leq R_T(\mathcal{H}) \leq O(\sqrt{d_L \cdot T \log T}).$
>
> Later, Alon et al. [2] showed that the precise regret is $R_T(\mathcal{H}) = \Theta(\sqrt{d_L \cdot T})$.
>
> For the _transductive_ agnostic online setting, we make the following novel observation:
>
> **Claim.** _Let $\mathcal{H}$ be a hypothesis class with VC dimension $d_V > 0$. Then the agnostic transductive regret for $\mathcal{H}$ with horizon $T$ is $R_T(\mathcal{H}) = \tilde{\Theta}(\sqrt{d_V\cdot T})$._
>
> An upper bound of $O(\sqrt{d_{V}\cdot T\log(T/d_{V})})$ follows from the Perles–Sauer–Shelah lemma + the multiplicative weights algorithm.
>
> **Proof sketch for lower bound.**
> Let $d=d_{V}$, and let $x_1,\dots,x_d$ be VC shattered by $\mathcal{H}$. Assume for simplicity that $T = kd$ for some $k \in \mathbb{N}$. The transductive adversary selects the sequence $x_1^1,\dots,x_1^k,x_2^1,\dots,x_2^k,\dots,x_d^1,\dots,x_d^k$, where $x_i^j = x_i$ for all $j \in [k]$ and $i \in [d]$. The adversary chooses all labels to be i.i.d. uniform random bits. By Khinchine’s Inequality (See Lemma 8.2 in [3]), for each $i \in [d]$ the expected gap between the majority label and the minority label in the subsequence $x_i^1,\dots,x_i^k$ is of order $\Omega(\sqrt{k})$. Hence, by an averaging argument, any learner will have excess error of order $\Omega(\sqrt{k})$ on roughly half of the $d$ subsequences $x_i^1,\dots,x_i^k$. This implies the desired lower bound of $\Omega(\sqrt{k} \cdot d) = \Omega(\sqrt{d\cdot T})$.
>
> Additionally:
>  * For trivial classes (e.g., a class that contains a single function), the regret is $0$.
>  * For classes with finite VC and finite Littlestone dimension, the regret is $\Theta(\sqrt{d_L \cdot T})$ (by [2]), namely, the $\log(T)$ factor from the upper bound for VC classes is removed.
>  * For classes with infinite VC, the regret is $T$.
>
> Hence, for non-trivial classes, finiteness of the VC dimension determines the asymptotic dependence of the regret on $T$ up to logarithmic factors.
>
> However, the regret for VC classes remains open because there is a gap of a $\sqrt{\log(T)}$ multiplicative factor between the best known upper and lower bounds. In other words, we don’t know whether there is just one case or several cases.
>
>
> _____
>
> [1] Shai Ben-David, Dávid Pál, and Shai Shalev-Shwartz. Agnostic Online Learning. COLT 2009.
>
> [2] Alon, N., Ben-Eliezer, O., Dagan, Y., Moran, S., Naor, M. and Yogev, E. Adversarial laws of large numbers and optimal regret in online classification. STOC 2021.
>
> [3] N. Cesa-Bianchi and G. Lugosi. Prediction, learning, and games. Cambridge University Press, 2006.

---

### Decision · Program_Chairs · 2023-09-21

**Decision:**

Accept (poster)

**Comment:**

The decision on this paper is difficult.

I will begin with the positives. First, the conceptual contribution from the trichotomy is valuable. From a technical standpoint, the main innovation is Theorem 3.1, and there is some novelty in the proof of Claim 3.4 as well.

There is also one (in my opinion) weakly positive point. The authors mention as a plus the didactic aspect. I recognize the importance of pedagogy, but I don’t see that as an argument for publication at a top venue (the teaching aspect is like a "cherry on top"). Of course, there is value in simplicity as it can make it easier for more complexity to be added.

Next, the downsides. Other than the results mentioned above, the proofs are straightforward. While there are results for the multiclass case, the technical innovations involved are described only in the appendix, and so it is difficult properly account for these innovations in evaluating the paper.

I have carefully avoided the discussion of the results for the agnostic setting, along with the DS dimension example. I will discuss those now; they are very important for the decision. The authors developed results for the agnostic case late in the review process (during discussion with the reviewers). These results complete the story of the paper and, if their addition to the paper is permissible at this late stage, significantly improve the outlook for this paper. Therefore, let us consider whether these new results should be considered. First, two reviewers are not in favor of considering these new results, while one reviewer is in favor, and (I believe) another also in favor. I will also weigh in. The results for the agnostic setting use standard techniques; hence, they are not that novel (this is a good thing, in this case, as including these results is more palatable). The DS dimension example also looks like a smaller addition (at the level of discussion). which makes me more more willing to go along with the inclusion of this example.

I think this work is interesting, and it carries enough strength for acceptance at NeurIPS.